

# Integrated risk assessment due to slope instabilities at the roadway network of Gipuzkoa, Basque Country

Olga Mavrouli [1], Jordi Corominas [2], Iñaki Ibarbia[3], Nahikari Alonso[4], Ioseba Jugo[3], Jon

Ruiz[4], Susana Luzuriaga[5] and José Antonio Navarro[5]

[1] Faculty of Geo-Information Science and Earth Observation (ITC), University of Twente, Enschede, Netherlands; previously at Department of Civil and Environmental Engineering, Technical University of Catalonia, BarcelonaTech (UPC)

[2] Department of Civil and Environmental Engineering, Technical University of Catalonia, BarcelonaTech (UPC)

[3] Ikerlur, S.L., Donostia

[4] LKS Ingeniería, S. Coop., Arrasate/Mondragon

[5] Diputación Foral de Gipuzkoa-Gipuzkoako Foru Aldundia, Donostia/San Sebastián

*Correspondence to*: o.c.mavrouli@utwente.nl & jordi.corominas@upc.edu

**Abstract.** Transportation corridors such as roadways are often subjected to both natural instability and cut slope failure, with substantial physical damage for the road infrastructure and threat to the circulating vehicles and passengers. In the early 2000s, the Gipuzkoa Regional Council of the Basque country in Spain, marked the need for assessing the risk related to the geotechnical hazards at its road network, in order to assess and monitor their

safety for the road users. The Quantitative Risk Assessment (QRA) was selected as a tool for comparing the risk for different hazards on an objective basis. Few examples of multi-hazard risk assessment along transportation corridors exist. The methodology presented here consists in the calculation of risk in terms of probability of failure and its respective consequences, and it was applied to 95 selected points of risk (PoR) of the entire road network managed by the Gipuzkoa Regional Council. The types of encountered slope instabilities which are treated are

rockfalls, retaining wall failures, slow moving landslides, and coastal erosion induced failures. The proposed methodology includes the calculation of the probability of failure for each hazard based on an extensive collection of field data and its association with the expected consequences. Instrumentation data from load cells for the anchored walls and inclinometers for the slow moving landslides were used. The expected road damage was assessed for each hazard level in terms of a fixed Unit Cost, UC. The results indicate that the risk can be comparable

for the different hazards. 12% of the PoR in the study area were found to be of very high risk.

## 1. Introduction

Transportation corridors such as roadways are often subjected to both natural instability and cut slope failure, with substantial physical damage for the road infrastructure and threat to the circulating vehicles and passengers. The



growing societal demand for road safety requires managing this risk, and places in high priority the identification of problematic areas to effectively manage the mitigation works.

Risk is most commonly conceptualized as the product of hazard, exposure, and vulnerability. Qualitative risk analysis for transportation corridors traditionally combines different levels of hazard and vulnerability to provide
the risk across the network (e.g. Pellicani et al. 2017).  Nevertheless, the interpretation of risk levels which are obtained qualitatively may vary for the different hazards (Eidsvig et al. 2017). The homogenization of the risk for multi-hazards remains a challenge because of the variability in the nature of soil or rock mass movement phenomena and the difference in the type and extent of the consequences. The comparison of different types of geotechnical risks in roadways, such as slope movements and retaining structures failure, requires bringing these
phenomena under a common denominator. Quantitative risk descriptors, as being objective expressions of the expected risk extent, may well serve for the homogenization of the risk levels for different hazards and types of exposed elements (persons, vehicles, infrastructure, and indirect economical loss). Common quantitative risk descriptors are the expected annual monetary loss, the probability of a given loss scenario, and the probability of one or more fatalities and others mentioned at Corominas et al. (2014).

One of the major limitations for the quantitative risk assessment in roadways is the great data demand that it implies. The hazard in terms of probability of an event of a given magnitude requires extensive data on the frequency and also magnitude (volume) of the events (Fell et al., 2008; Jaiswal et al. 2010). Most commonly, landslide inventories are required (Dai et al., 2002; Ferlisi et al. 2012), although in most cases they are scarce. Highway and traffic administration authorities are potential data providers (Hungr et al., 1999), however complete
and reliable maintenance records are rarely kept and made available. Alternative methods to overcome the scarcity of empirical data are provided at Corominas et al. (2014), and they are based on geomechanical or indirect approaches. They associate the occurrence of events with the temporal occurrence of their triggering factors, such as  the return period of a rainfall of a given intensity. On the other hand, the calculation of the consequences in terms of realistic expected costs is a challenge for a purely quantitative risk assessment, as the amount of repair or
insurance expenses fluctuates greatly depending on the type and extent of the damage, on top of the indirect costs related to traffic interruptions, detours and further loss related to traffic accidents.  Due to these limitations, few purely quantitative multi-hazard risk assessments for roadways exist in the literature.

An extensive review of highway slope instability risk assessment systems is provided by Pantelidis (2011), including several qualitative and semi-quantitative methods (here by semi-quantitative we refer to the
methodologies that assess the hazard in terms of numerical scores). A well-known example of semi-quantitative methods is the Rockfall Hazard Rating System (Pierson and Van Vickle, 1993) recommended by the FHWA (Federal Highway Administration of the United States), which was later adapted by Budetta (2004), specifically for rockfall risk along roads.  The pure quantitative risk assessment (QRA) however consists in the hazard assessment in terms of probability of failure/occurrence of an event of a given magnitude multiplied with its
respective consequences (Fell et al. 2008), which is not treated by semi-quantitative methods.

Hungr et al. (1999) quantified the rockfall risk along roadways at the British Columbia, after deriving magnitude-cumulative frequency curves. Bunce et al. (1997) were amongst the first to use the roadway damage in order to assess the rock fall frequency. Remondo et al. (2008) proposed a method for the quantification of the damage at the Gipuzkoa road network where losses were calculated on the basis of past damage records and taking into


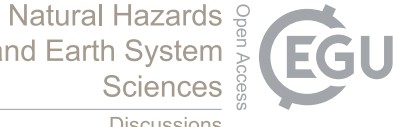


account budgets for road and railway track repairs. Similarly, Zêzere et al. (2008) assessed the direct risk from translational, rotational and shallow landslides in the north of Lisbon, Portugal, employing road reconstruction costs within a Geographical Information System. Ferlisi et al. (2012) using the fundamental risk equation provided by Fell et al. (2008) calculated the annual probability of one or more fatalities by rockfalls in the Amalfi coastal

road, Southern Italy, and Michoud et al. (2012) presented an example from the Swiss Alps. Jaiswal et al.(2010) applied a risk model for debris slides, where the temporal probability was indirectly obtained by the return period of the triggering rainfalls and the road vulnerability was assessed in function of the road location and expected debris magnitude. Ferrero and Migliazza (2013) made a first attempt to incorporate the efficiency of protection measures into the risk assessment (Nicolet et al. 2016). Still, when it comes to the practical application of

quantitative risk analysis to linear infrastructures, several challenges exist.

One of the most important is the assessment of the expected magnitude-frequency relations and of the annual probability of occurrence for a hazard of a certain magnitude/intensity, in particular where past event inventories are incomplete or missing. As afore-mentioned, alternative methods have been suggested to this end, however in most cases their application is limited to site specific. There is a scarcity of cost-efficient, quick, simple and easy

enough methodologies, to be applied to extensive road networks, using as input the evidences that can be found in the vicinity of the transportation corridors, field inspections or instrumentation. Given those limitations, the determination of landslide magnitude-frequency data requirements and its specifications, within a suitable and feasible framework for transportation corridors, remains an issue.

Assessing the condition of assets such as road pavements and protection infrastructure (in this case retaining walls)
allows for monitoring operational efficiency, planning future maintenance and rehabilitation activities and controlling costs, through condition forecasting models (Gharaibeh and Lindholm, 2014). Although models for predicting pavement deterioration under usual stress conditions have been used for more than three decades now, literature is lacking prediction models for disruptive slope instability events. Similarly, simple yet functional and (semi)quantitative-based empirical models for the condition assessment of retaining walls are scarce.

Diverse hazards types require different descriptors for predicting asset condition. In quantitative multi-hazard risk assessment, the use of all descriptors should produce comparable results at a common and meaningful (commonly financial) scale. This requires, in each case, adequate criteria and thresholds for the establishment of hazard classes, to associate with vulnerability levels and costs (Schmidt et al. 2011; Kappes et al. 2012). Very few examples of roadway vulnerability exist in the literature (e.g. Mansour et al. 2011; Eidsvig et al. 2017). Their applicability or
adjustment to other case-studies is a topic for further research.

The work that is presented here aims at filling in these gaps for the development of a comprehensive procedure including suitable data collection, hazard and vulnerability assessment and their integration into risk calculations. Up to date and to the authors' knowledge there are scarce integrated approaches for multi-hazard quantitative risk analysis at transportation routes, at site-specific and local scale. The starting point was the need for a risk
assessment system for a specific area, and all approaches discussed here for confronting these issues are strongly related to the local characteristics of the study-area and the available documentation and instrumentation.

In particular, in the early 2000s, the Gipuzkoa Regional Council of the Basque country in Spain, marked the need for assessing the risk related to the geotechnical hazards at its road network, in order to assess and monitor their safety for the road users. The main objective has been the identification of the most problematic areas where

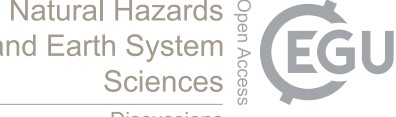



mitigation measures should be prioritized. In that specific road network, a variety of geotechnical hazards coexist, which are relevant to both cut and natural slope instabilities, considering as well coastal erosion, and including the potential failure of retaining walls. A quantitative risk analysis approach was proposed.

The methodology presented here was developed with the objective to compare the risk levels, for a variety of
5    elements comprising roads and retaining walls, using a common unique criterion for their evaluation. It consists in the quantitative risk assessment (QRA) in terms of probability of failure and its respective consequences, at about 100 selected points of risk (PoR) of the entire road network managed by the Gipuzkoa Regional Council (Figure 1). The types of encountered slope instabilities which are treated in this manuscript are rockfalls, retaining walls, slow moving landslides, and coastal erosion induced failures. Further geotechnical risks in the area include
10   debris flows, instability of embankments and brittle slope failures, but their assessment is not included here.

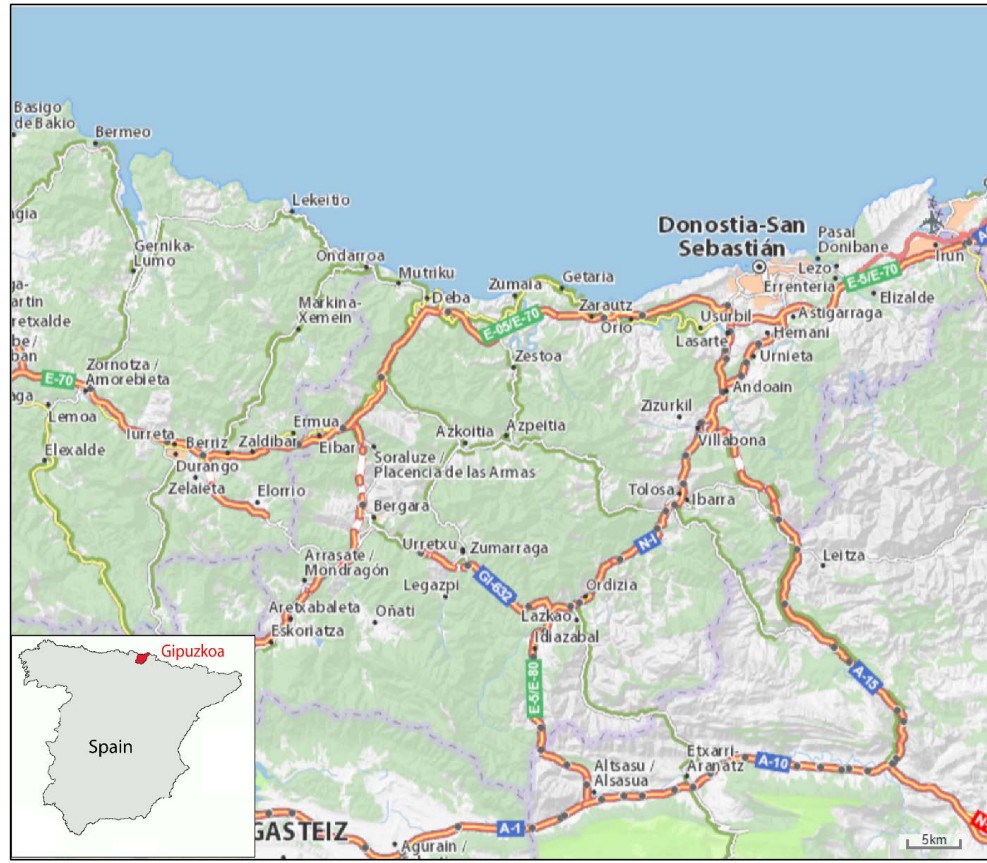

**Figure 1: Road network managed by the Gipuzkoa Regional Council (map source ViaMichelin website at https://www.viamichelin.com/web/Maps, last accessed in September 2018).**

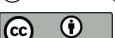



## 2. Study area and data availability

The study area is the road network managed by the Gipuzkoa Road Authority, in the Basque Country, Spain (Figure 1). It consists of four networks: highways, primary, county, and local roads. Their difference lies in their capacity and function as connecting corridors between major and/or minor urban/rural nuclei. In that region, the layout of the road network has been spatially constrained since its design by its characteristically intense morphological relief. Soil infills or excavations and important constructions such as retaining walls are required to protect the road users and the road infrastructure against soil/rock mass movements and instabilities. An important fraction of the retaining walls is anchored walls.

From a geological point of view, the Gipuzkoa province is part of the Basque-Cantabrian basin (Barnolas and Pujalte, 2004). More specifically, it is the segment connecting the Pyrenees and the Cantabrian mountains to the West (Tugend et al. 2014). It experienced normal faulting and high subsidence rates during the Cretaceous and was inverted during Tertiary compression related to the Alpine orogeny (Gómez et al. 2002). The outcropping rocks cover a wide temporal record, from the Upper Paleozoic to the Quaternary. However, there is no representation of the materials belonging to the period between the Lower-Middle Tertiary (Oligocene) and Early Pleistocene. The geology of the study area is well synthetized by Ábalos (2016) (Figure 2).

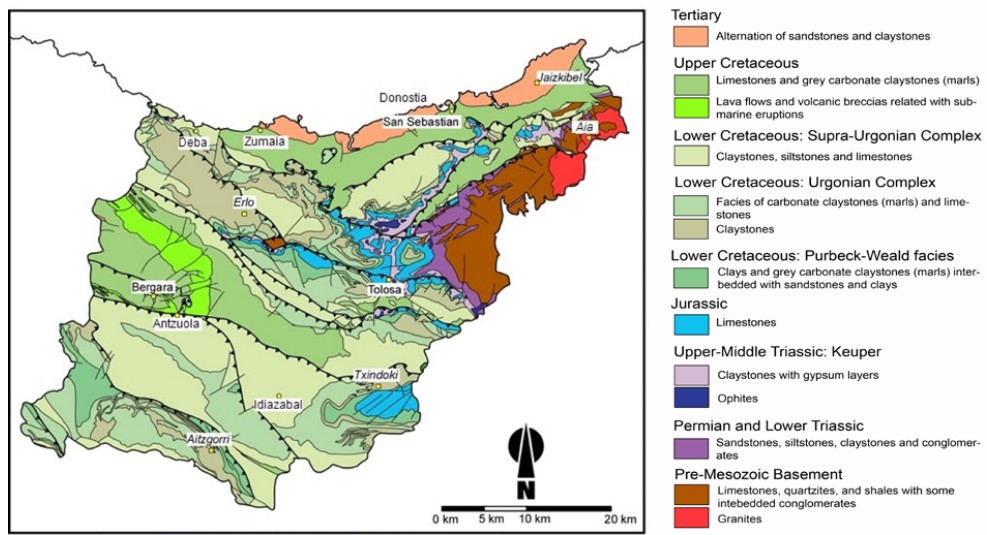

**Figure 2: Geological sketch of Gipuzkoa province (modified from Ábalos, 2016).**

Two well differentiated geological sectors can be distinguished. The first one extends over the NE corner of Gipuzkoa, in which the Paleozoic rocks (granitoid rocks and limestone layers overlaid by quartzites and shales with some interbedded conglomerate layers) and Triassic rocks (sandtones, siltstones, claystones and conglomerates) are predominant. The rest of the Gipuzkoa province is composed of thick sedimentary assemblages, mainly Cretaceous and Tertiary (Figure 2). The outcropping lithological units are diverse. Jurassic rocks are usually composed of carbonate rocks of marine platform facies. Several stratigraphic units are found in the Lower Cretaceous formations. The Weald complex is composed of continental and freshwater sedimentary



rocks. The Urgonian complex is formed by reefal limestones; the Supra-Urgonian Complex exhibits a considerable internal complexity , mostly clayey limestones, siltstones, and argillites. The Upper Cretaceous is characterized by Flysch alternations and Tertiary flyschoid sandstones. In general, the sedimentary rocks were formed in a great diversity of depositional environments, although they mostly correspond to marine environments, with
intercalations of continental or transitional facies.

As the resistance to erosion depends on the lithology, strong materials such as limestones, conglomerates, and sandstones stand out, constituting the topographic reliefs, versus the more easily erodible areas of soft materials, such as clays and siltstones.

Concerning tectonics, all the aforementioned materials appear folded and fractured as a result of the actions during
the two orogenic phases. The oldest one, corresponding to the hercynian orogeny, affected the Paleozoic rocks while the more recent alpine orogeny has been the responsible for the general uplift of the Pyrenean Mountain Range. During the uplift, the Mesozoic sedimentary cover detached from the Paleozoic basement, in favour of the most deformable lithological units (e.g. Keuper), and created a complex structure of folds faults and diapirs. Both the folds and the main fractures have alignments oriented in the direction NE-SE and NW-SE, which in the eastern
part of the Gipuzkoa province form an arch that bounds the Paleozoic reliefs.

The Quaternary rock deposits pertain to residual soils, originating from the disintegration, weathering or dissolution of the underlying rock mass, without having undergone transport (colluvium, scree deposits on the foot of steep slopes), and alluvial soils.

Since 2002, the Department of Mobility and Road Infrastructure of Gipuzkoa has developed and installed an
extensive system for the surveillance and monitoring of the anchored retaining walls with hydraulic or vibrating wire load cells, placed over the most representative and critical retaining structures and slopes. The monitoring system is completed with further instrumentation of the slopes and embankments, comprising piezometers, inclinometers measuring quarterly to semi-annual displacement and extensometers, controlling, on a continuous basis, 100 Points of Risk (PoR). 95 of them involve rockfalls, potentially unstable retaining walls, slow moving
landslides and potentially unstable sea walls with overlaying roads. In spite of the instrumentation data available in the area, the comparison of the risk levels at the different PoR and the establishment of a methodology for the homogenous multi-hazard quantitative risk assessment have remained unresolved until the proposal of the methodology that is presented here.

Five road types based on the Average Daily Traffic (ADT) can be distinguished (Table 1), for which the
geotechnical hazard situations vary. For the sake of brevity, only the four aforementioned types of instability are

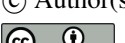



treated here. Typical examples of them in the study area are shown in the Application section. In particular, the different hazard situations can be summarized as follows:

- rockfalls with or without mitigation measures;
- anchored walls partially or totally instrumented with inclinometers and/or load cells inspected on a continuous or annual basis;
- instrumented slow moving landslides with available geotechnical study;
- sea walls for road construction or protection of roads against sea erosion, with available geotechnical study and documented periodical revisions.

**Table 1: Occurrence of geotechnical hazard types at different road types (data source: Department of Mobility and Road Infrastructure of Gipuzkoa).**

| Road Type | ADT | Networks | Anchored walls | Rockfalls | Slow moving landslides | Sea walls |
|---|---|---|---|---|---|---|
| 1 | >25,000 | Highway / Primary | x | | x | |
| 2 | 15,000-25,000 | Highway / Primary | x | x | x | |
| 3 | 5,000-15,000 | Highway/Primary / County | x | x | x | x |
| 4 | 1,000-5,000 | Primary / County / Local | x | x | x | |
| 5 | <1,000 | County / Local | | x | x | |

ADT: Average Daily Traffic

Rainfall data has been available for the study site by the Meteorological Agency of the Basque country. Further data has been collected after periodical field inspections and from the monitoring network, the type of which differs according to the hazard. This is detailed in the following section. Periodic inspections have been on-going up to date.

**3. General Methodology for the Risk Assessment**

The objective of the general methodology that is presented here is to compare, on a common basis, the risk at the different PoR. The risk components that are used are the hazard and the consequences. For the calculation of the risk, the methodology takes into account the repair of damage in order to restore normal traffic. This expresses the direct risk for the road. Indirect loss such the economic impact of the road blockage and detours are not described here. The risk quantification in terms of monetary loss requires calculating repair costs for different damages for each hazard, as described in Table 2.

The hazard is expressed in terms of annual probability of failure of a natural/cut slope or retaining wall, of a given magnitude j. For dormant landslides, it is the probability of a sudden reactivation. Magnitude (volume) or intensity (velocity) descriptors were defined for each hazard. The consequences include costs related to removal of rubble, repair/replacement of the pavement, scaling of the slopes (the removal of loose non detached rock or debris), and slope stabilization. The cost is evaluated in multiples of a Unit Cost, UC, set at 1,000 €



**Table 2: Hazards, instability mechanisms, and principal consequences**

| a/a | Symbol* | Instability Mechanism | Principal consequences |
|---|---|---|---|
| 1 | RF | Rockfall | Pavement occupied |
| 2 | RS | Failure of anchored retaining structures | Failure of structure and pavement occupied |
| 3 | SL | Slow moving landslide | Deformation and/or road failure |
| 4 | SW | Failure of sea wall | Deformation and/or road failure |

If more than one types of hazards are present on a given PoR, the total risk is the sum of risks.

The overall methodology for the quantification of the risk consists in the general application of Eq. 1.

$$R_T = \sum_j^k P_{rk} * C_k \, , \tag{1}$$

Where:

$R_T$: Average annual risk in terms of UC per year

j: Magnitude (volume) or intensity (velocity) class

k: Hazardous event type (rockfall; failure of an anchored retaining wall; slow moving landslide; failure of a sea wall)

$P_r$: Annual probability/frequency of occurrence of a failure/rupture of magnitude j. For slow moving landslides, it refers to the probability of acceleration of a landslide with a given level of intensity (velocity).

$C_k$: Consequences of the failure/rupture caused by a hazardous k-type event, of magnitude j in terms of (as multiples of) the UC (set at 1,000 €).

The magnitude classes of the adverse events are established empirically based on the observed consequences (road damage) and the average cost of the remedial measures typically undertaken (see appendix).

For each hazard further assumptions are made and steps are followed for the evaluation of the components of Eq.
(1), which are described in the following sections.

**3.1 Rockfalls (RF)**

Rockfalls are a major threat at the roads of Gipuzkoa. The rockfall hazard magnitude is classified according to the volume of the detached mass. A frequency or probability of occurrence is attributed to each volume class and the
extent of disruption of the transportation corridor is determined.

For the frequency-magnitude relation, a catalogue of events is available only for limited sections of the road network. An inventory of rockfalls was compiled for the road N-634 connecting Zarautz and Getaria, based on highway administration data and press sources. According to the recorded historical events, six (from A to F) volume ranges were considered (Table 3). For the sections of the road N-634, the frequency was then be assessed
as the number of events divided by the number of observation years.





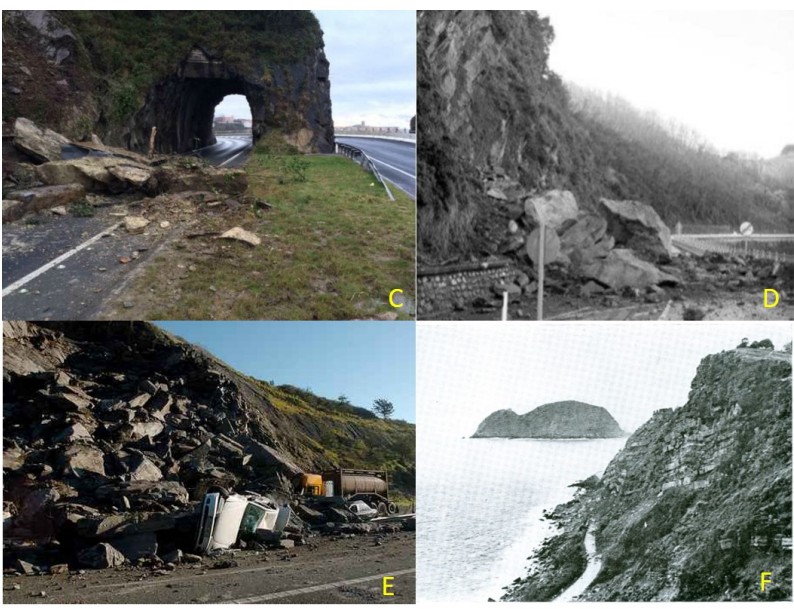

**Figure 3: Examples of rockfall magnitudes (C, D, E, and F) in the study area, in relation to the classes of Table 3 (source: photo D, E: Javi Colmenero; F: Servicio Geológico de Obras Públicas)**

**Table 3: Magnitude classes and respective annual frequency for the road N-634 connecting Zarautz and Getaria.**

| Class | Volume (m³) | # events | Period | Time span (years) | events/yr) |
|---|---|---|---|---|---|
| A | <1 | 104 | 1994-2009 | 15 | 6.933 |
| B | 1-5 | 91 | 1994-2009 | 15 | 6.067 |
| C | 5-50 | 24 | 1994-2009 | 15 | 1.600 |
| D | 50-500 | 5 | 1960-2009 | 49 | 0.102 |
| E | 500-5,000 | 1 | 1884-2009 | 125 | 0.008 |
| F | >5,000 | 1 | | >330 | <0.003 |

10   Although the same amount of information is not available for all the PoR, numerous in situ inspections at the PoR have provided extensive topographical and geological data, which can be used for estimating the frequency at the rest of the sections. To this aim, a relation was established between the frequency of events and certain geological parameters of the slopes, for 18 representative example sections on the road N-634. The same relation was then applied to the PoR without rockfall inventories. The main assumption is the equivalence of the frequency-

15   magnitude relation at slopes with similar geostructural characteristics, number of recent scars, slope height and block size. The suggested frequency indicators are the joint persistence ($I_P$), the scar density ($I_{DC}$), differential erosion ($I_E$), the number of potentially unstable rock masses ($I_{FP}$), and the Slope Mass Rating Index ($I_{SMR}$). These indicators are calculated as follows:





- The persistence of discontinuities in the rock mass, $I_P$, is assumed high when planes of several tens of meters can be observed on the slope face; moderate when of some meters; and low when it is sub-metric. The stratification is taken into consideration as well. Higher persistence of discontinuities results in the existence of more planes permitting the block detachment from the slope face.

5   - The scar density, $I_{DC}$, is calculated as the ratio of the number of recent scars to the slope height. Scars can be noticed as areas with a different colour (often reddish) from the rest of the slope face. Greater number of recent scars indicates higher and more frequent activity.

- Differential erosion, $I_E$, can be present or absent and can be observed at rock masses constituted by materials of distinctive strength. Differential erosion leads to loss of support for the overlying rock mass.

10   - The number of potential rockfalls, $I_{FP}$, refers to the number of potentially unstable rock masses as observed by in situ inspections. This number is collected distinguishing between magnitude classes.

- The Index Slope Mass Rating $I_{SMR}$ is the value of SMR, as proposed by Romana (1991).

**Table 4: Scoring of the indicators used for the rockfall frequency assessment.**

| Scores | 0 | 1 | 2 |
|---|---|---|---|
| Frequency Indices | | | |
| $I_P$ (Joint persistence)* | Low | Moderate | High |
| $I_{DC}$ (Density scars)** | <0.1 | 0.1-0.3 | >0.3 |
| $I_E$ (Differential Erosion)*** | No | Yes | |
| $I_{FP}$(Number of points with potential rockfalls)**** | <2 | 3-10 | >11 |
| $I_{SMR}$ (SMR)***** | >80 | 40-80 | 0-40 |

Each indicator scores 0, 1 or 2 points applying the criteria of Table 4. The indicator scores are summed up to provide the frequency index $I_F$ according to Eq. 2. The $I_F$ was first calculated at the cut slopes with an available inventory of rockfall events in order to establish a relationship between the $I_F$ and the frequency of events of all magnitudes. After calibration, the thresholds of this relationship were established as shown in Table 5.

$$I_F = I_P + I_{DC} + I_E + I_{FP} + I_{SMR} , \hspace{3cm} (2)$$

**Table 5: Annual frequency $f_a$ for different $I_F$ values.**

| $I_F$ | Annual frequency $f_a$ (events/yr) |
|---|---|
| ≥8 | ≥3 |
| 6-8 | $1≤f_a≤2$ |
| 5-6 | $0.2≤f_a≤1$ |
| 3-5 | $0.1<f_a≤0.2$ |
| 0-3 | ≤0.1 |



To assess the expected frequency $F_r$ of a given rock size, the total expected number of events is distributed over the rockfall volume classes, as observed in situ. If this datum is lacking, the proportion of a modal size of blocks multiplied by the total annual frequency is suggested instead.

The implementation of measures of stabilization (bolts, anchors), retention (nets and/or gunite), or protection

5 (barriers, galleries), partially or entirely, reduces the annual frequency of rock blocks reaching the roadway. To account for this, a corrected reduced frequency $F_{rc}$ is proposed to be used, upon a correction factor n (Table 6), according to Eq. 3:

$$F_{rc} = \frac{F_r}{10^n} \ , \tag{3}$$

where,

$F_{rc}$: annual corrected frequency

$F_r$ : annual frequency before correction

N: correction factor given by Table 7.

**Table 6: Correction factor for different protection measures, to be applied on the annual frequency according to Eq. 3, for each magnitude.**

| Magnitude (m³) | A | B | C | D | E | F |
|---|---|---|---|---|---|---|
| Volume (m³) | <1 | 1-5 | 5-50 | 50-500 | 500-5,000 | >5,000 |
| Gunite, bolts (effective and extense) | 0.5 | 0.25 | 0 | 0 | 0 | 0 |
| Gunite, bolts (partial) | 0.25 | 0.1 | 0 | 0 | 0 | 0 |
| Shotcrete with bolts | 1 | 0.5 | 0 | 0 | 0 | 0 |
| Cable nets | 0.25 | 0 | 0 | 0 | 0 | 0 |
| Triple torsion nets | 0.5 | 0 | 0 | 0 | 0 | 0 |
| Low rigid foot barrier | 0.25 | 0 | 0 | 0 | 0 | 0 |
| High rigid foot barrier | 1 | 0.5 | 0 | 0 | 0 | 0 |
| High rigid foot barrier (partial) | 1 | 0.5 | 0 | 0 | 0 | 0 |
| Ditch width <5 m | 0.2 | 0 | 0 | 0 | 0 | 0 |
| Ditch width <5 m with vegetation | 0.5 | 0.1 | 0 | 0 | 0 | 0 |
| Ditch width >10 m | 0.5 | 0.25 | 0 | 0 | 0 | 0 |
| Ditch width > 15 m | 1.5 | 1 | 0 | 0 | 0 | 0 |
| Flexible Barriers <2000KJ | 3 | 1 | 0 | 0 | 0 | 0 |
| Flexible Barriers >2000KJ | 3 | 2 | 1 | 0 | 0 | 0 |
| Gallery | 4 | 3.5 | 3 | 2 | 1 | 0 |

The thresholds of Table 4 for the scoring of indicator, the $I_F$ values and the annual frequency as indicated in Table

20 5, were obtained after a trial and error iterative calibration procedure, so as to optimize the matching of the results with the observed frequency from real events at natural slopes (the latter marked as 1-2 events per year, or 1 event every 1-5 years, or 1 event every 5-10 years). The calibration yields results which are overestimation at maximum 2 events per year, in 5 section, while the rest of the results are compatible. At a second stage, further calibration was performed considering the protection measures, which yields the correction factors of Table 6.





The risk at each point is then calculated for each PoR by the general Eq. (1). For this, the consequences are assessed per rockfall magnitude classes as indicated in the appendix. The five magnitude classes also shown in Table 3 were defined judgmentally, based on the principal consequences and disruption of the road, as observed from previous rockfalls and road maintenance interventions. In Table 14 (Appendix) the principal consequences and disruptions are shown for each magnitude class, as the respective actions which are used as a guide for the establishment of the costs in terms of UC.

**3.2 Failure of retaining structures (RS)**

One of the objectives of this work was the analysis of the risk related to the failure of anchored reinforced concrete walls. The uncertainties characterizing the structural design parameters and the terrain resistance are substantial, thus the safety level of these structures cannot be precisely assessed. The probability of failure is considered instead.

The hazard level associated to the anchored retaining walls is evaluated on the basis of a heuristic hazard index, HI. The evaluation consists in a modification of the Methodology for the Revision of Anchors, developed by the company Euroestudios in 2004. According to it, the HI for the retaining structures is equal to the average of the scores assigned to three components (Eq. 5). These components and their scores are presented in Table 7 and they are: the safety factor of the wall, the anchorage design (DA), and the project and construction quality (PQ). The scoring for each component range between 1 and 5. Moreover, to calculate the scoring of the index DA, the average value from 3 parameters is considered according to Eq. 4: % working load/ultimate load ratio (UL), grout length per ten tons load (GL), and the anchoring ground (AG).

**Table 7: Scores the hazard index HI components, for the failure of retaining walls.**

| Parameter | | Value | | | | |
|---|---|---|---|---|---|---|
| | | 1 | 2 | 3 | 4 | 5 |
| Safety Factor (SF) | | >2 | 1.55-2.0 | 1.45-1.55 | 1.30-1.45 | <1.30 |
| Anchor Design (DA) | Working load (% Ultimate load) UL | <55 | | 55-65 | | >65 |
| | Grout Length GL | >1.2m/10t | | 0.8-1.2m/10t | | <0.8m/10t |
| | Anchoring ground AG | Sound Rock | | Mixture | | Weathered soil/rock |
| Project and construction (PQ) | Available data for anchors | yes | | no | | no |
| | Technical assistance during construction | yes | | yes | | no |

The hazard index (HI) is obtained using the following Eq. (4) and (5):

$$DA = \frac{UL+GL+AG}{3},$$ (4)





$$HI = \frac{SF+DA+PQ}{3},$$ (5)

The thresholds and the scoring of the parameters was made based on expert judgement, increasing the hazard index
when the function of the anchors is critical or uncertain, and decreasing it when loading conditions and supports,
as well as when good practices in construction can guarantee a good function of it. In Equations (4) and (5), for
simplicity all factors are equally weighted.

Silva et al. (2008) have associated the safety of factor of engineered slopes for a given project category with the
probability of failure. For this, the structures are classified according to the level of engineering design. Category
I comprises engineered slope which were designed, constructed and managed using the most advanced state of the
art knowledge. Category II includes constructions with normal standards. Category III are constructions without
specific design that do not follow standards, and Category IV those ones with a poor or completely missing
engineering knowledge basis. As the HI is an adaptation of the SF that takes into consideration additional criteria
for the level of functioning of the structure, a similar relation between annual probability of failure of the anchored
retaining walls can be established as well. The relation is presented in Figure 4 and Table 8.

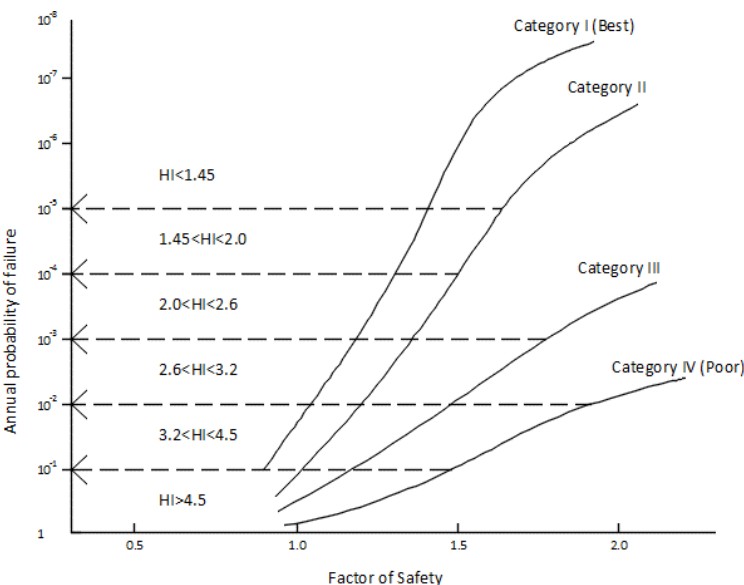

**Figure 4: Classification of the engineered slopes and relation to the annual probability of failure (modified**
**from Silva et al. 2008).**



**Table 8: Annual probability of failure for HI values, according to Figure 4.**

| Annual probability of failure (Pr) | Hazard Index (HI) |
|---|---|
| 0.1<P | HI >4.5 |
| 0.01<P≤0.1 | 3.2<HI≤4.5 |
| 0.001≤P≤0.01 | 2.6<HI≤3.2 |
| 0.0001≤P≤0.001 | 2.0<HI≤2.6 |
| 0.00001≤P≤0.0001 | 1.45<HI≤ 2.0 |
| P≤0.00001 | HI≤1.45 |

The proper design and construction of the anchored retaining walls should be reflected in the absence of deformations and anchor overloading. Increased pressure at the load cells, deformations, cracks and wall tilting are interpreted as instability indicators. In that case the annual probability of failure $P_r$ should be increased. Thus, factors of increase are added to the initial value of the HI, as shown in Table 9. The availability of this information implies that periodic and detailed wall inspections are carried out.

**Table 9: Increase factor of the HI according to instability indicators at the retaining walls.**

| HI Increase | Increase of the service load | Deformation of the retaining wall and or/terrain |
|---|---|---|
| 1 | Pressure increase of <65% of ultimate load | Deformation <3mm/a |
|  | External factors (groundwater table changes, erosion,…) | Cumulative deformation >30% of the maximum allowable deformation of the concrete |
| 2 | Pressure increase of >65% of ultimate load | Deformation >3mm/a |
|  |  | Cumulative deformation >60% of the maximum allowable deformation of the concrete |
|  |  | Presence of cracks, tilting, etc |
|  |  | Non instrumented retaining wall |

For measuring the consequences, we distinguished between the failure of small retaining walls, retaining walls shorter than 6m, and higher than 6m (Table 14-Appendix). The length of the affected road section, considering the spreading of the debris, was empirically fixed as the triple of the wall height at the section.

**3.3 Slow moving landslides (SL)**

The involved landslides at the PoR have a persistent creeping movement, which in a worst-case scenario can lead to a sudden acceleration. The clay materials in the study area have a viscous behaviour, characterized by resistance increase as the movement rate increases. High rainfall precipitations often result in landslide reactivation, with centimetre displacements. As this analysis involves active landslides or landslides with episodic reactivations, the challenge here was to assess the probability of a given damage level, in function of the landslide movement rate.



Mansour et al. (2011) indicated that a relation may be established between the damage expected from slow-moving slides to roads, versus the displacement rate. They proposed ranges for the annual displacement rate leading to different extents of damage, which are 0-10, 10-100, 100-160, 160-1,600, and >1,600 mm/year corresponding, respectively, to limited, minor, moderate, severe damage or total road destruction. They also provided a description for each damage extent with respect to the type and frequency of the actions to be taken for its repair. The aforementioned ranges do not distinguish between horizontal and vertical deformations, on one hand, and on the other, they are based on the assumption that continuous, almost constant, movement takes place.

Instead, our experience in Gipuzkoa shows that landslide reactivation is episodic and can be sudden, producing deformations even in short periods, such as few days or weeks. Although those deformations are usually of centrimetric order, they can cause cracks, bumps and puddles on the pavement, and jeopardize road traffic safety. For this reason, more restrictive criteria that those proposed by Mansour et al. (2011) were applied. They relate the maximum observed horizontal landslide velocity and the annual horizontal displacement rate to the annual probability of exceeding a damage level. The following paragraphs describe how they were established.

In the case study, horizontal velocity data is available from the inclinometer measurements. We inspected the in-situ damage, in order to establish the relation between the velocity and the road damage. Four levels were identified, as shown in Figure 5 and described qualitatively in the following:

- Damage level A (landslides of low intensity): There are rarely cracks and deformations. No speed reduction is required while driving. Crack sealing and resurfacing is periodically performed for periods longer than 1 year.

- Damage level B (landslides of moderate intensity): Damage includes pavement deformation and cracks and/or roadside destruction, without affecting the functioning of the road. If proper signalling is present, the chance of an accident is very low. Traffic conditions can be normally maintained if regular sealing and resurfacing is performed, until a more permanent solution.

- Damage level C (landslides of high intensity): Pavement deformation is substantial, including the presence of steps and puddles, and partial rupture of the platform affecting partly or entirely traffic lanes and/or the roadside. The normal functioning of the road is disturbed. Traffic is not interrupted, although restrictions such as alternative pass and traffic lights regulation are required. Accidents might occur if the vehicles enter the affected zone, without previous signalization. The normalization of the traffic conditions requires slope stabilization and road repair.

- Damage level D (landslides of very high intensity): Displacements are of the same order of magnitude as for level C, however with the vertical component prevailing giving place to complete destruction and loss of the pavement continuity. This situation is typical for roads situated on the crest or on the boundaries of landslides. The operation of the road is seriously affected. There is a high likelihood of an accident due to vehicles falling into the generated depressions. Important stabilization works and reconstruction of the road are required.





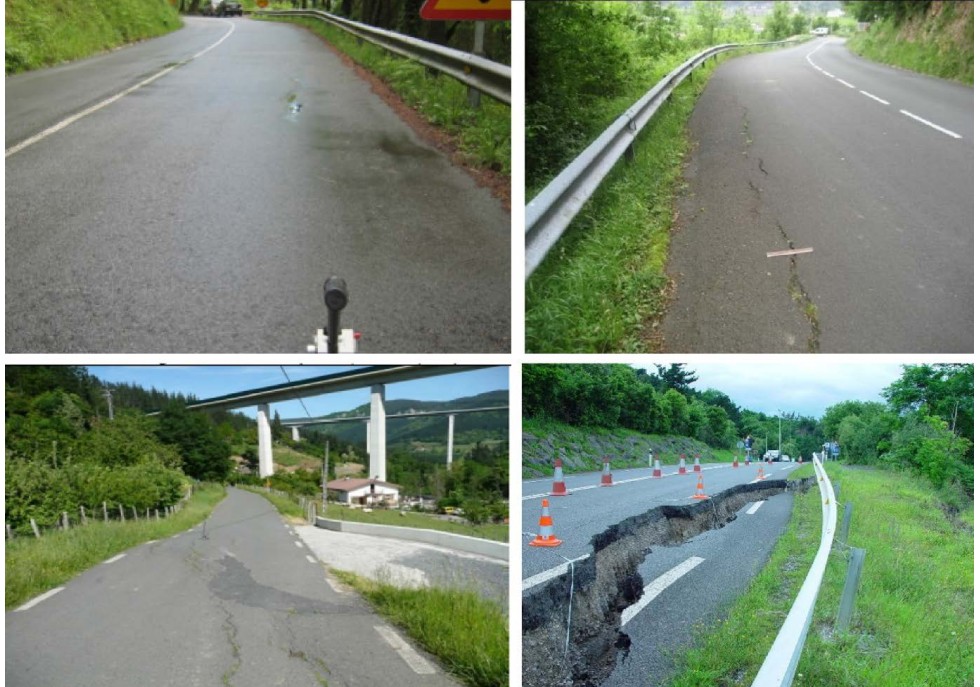

**Figure 5: The four damage levels identified at the roads of Gipuzkoa: (top-left) No/Slight damage, (top-right) Moderate damage, (down-left) Severe damage, and (down-right) Partial/total destruction.**

To establish damage proxies for the calculation of the expected consequences we correlated the observed road damage with the indications of the inclinometers at 24 reference PoR. The correlation was found to be positive, as, in most cases, increasing displacements were associated with more severe damage. In particular, the maximum horizontal monthly velocity and the cumulative displacement were used as the two proxies for the damage. For the establishment of the thresholds that relate the damage level with the terrain displacements, we tried to maximize

the right predictions (when the observed damage level is the same with the calculated damage level), and at the same time to achieve a balance between damage underestimation and overestimation. The proposed thresholds are summarized in Table 10. Out of the 24 PoR, 20 yield right predictions, 1 presents damage overestimation and 3 damage underestimation (out of which one inclinometer measures has low reliability). Partial/total destruction occurs when the criteria for severe damage are fulfilled plus either one of two further criteria: the road is located

inside the landslide scarp or a shear crack has being formed (the scarp manifests itself as a semi-circular form on the pavement).

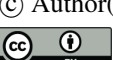



**Table 10: Damage criteria in function of landslide velocity (MHDR: maximum horizontal displacement rate; ACDR: annual cumulative horizontal displacement).**

| Damage levels | MHDR (mm/month) | ACDR (mm/a) | Landslide intensity | Further criteria |
|---|---|---|---|---|
| A: No / slight damage | <3 | <30 | low | |
| B: Moderate damage | 3-10 | 30-100 | moderate | |
| C: Severe damage | >10 | >100 | high | |
| D: Partial/total destruction | >10 | >100 | very high | Road inside landslide scarp and/or shear crack |

For the slow moving landslides, the hazard was expressed in terms of temporal probability of reactivation with an intensity exceeding a given level of damage. To assess the temporal probability we first distinguished between the landslides that are responsive to intense rainfall precipitations, from those for which a clear relation between rainfall and reactivation cannot be established. For each case, typical movement patterns, were observed by the inclinometer measurements. Four patterns were identified depending on the maximum monthly velocities. For

each type (responsive and not responsive) and each pattern O, X, Y or Z as described in the following, the probability of reactivation with an intensity leading to low (A), moderate (B) and high (C) or very high (D) damage is determined.

As most landslides in the study area are creeping undergoing continuous small deformations, the probability of low damage is always high (P∼1). For some of them, acceleration takes place for intense or long rainfall periods.

Moderate, high or very high damage might then occur with an annual probability equal or lower than 1. Its assessment of which is herein based on the return period of two major extreme rainfall events in the area.

The two extreme events were recorded during the observation period covered by the monitoring network of the Gipuzkoa Road Authority and they are: (1) the rainfall events of 4 to 7 November 2011 that were of high intensity and short duration. The rain recorded on the 6th of November 2011 was 185 mm, which according to the Water

Management Agency of the Basque Country, corresponds to a return period of the order of 100 years; (2) the rainy period of January-February 2013 that was characterized by moderate to low daily intensity but of long duration, with cumulative precipitation measurements that exceeded the maximums of the reference period 1971-2000, and a return period of over 100 years according the Euskalmet (Basque Meteorological Agency).

Accordingly, it can be assumed that the annual probability of reactivation or sudden acceleration of the

instrumented landslides that have not experienced deformations during the two afore-mentioned events is lower than $P = 0.01$ ($\sim \frac{1}{100}$). Using this probability as a reference, the reactivation probability for the different patterns and types is defined empirically, according to the observed number of peak month velocities on the inclinometer measurements.





For slow moving landslides which are responsive to rainfall events, we distinguish between four movement patterns (Figure 6).

- Pattern O: Landslides which are inactive or extremely slow, which have not experienced deformations neither in extreme nor in common precipitation conditions. The annual probability of a reactivation is P=0.01.

- Pattern X : Landslides with a high probability of reactivation and low intensity, characterized by displacement rates lower than 2-3 mm/month and cumulative displacements lower than 30 mm (Figure 6, upper-left). They have not experienced significant accelerations for the two afore-mentioned extreme events. Low damage A is mostly expected.

- Pattern Y: Landslides with a high probability of reactivation and low/moderate intensity, characterized by displacement rates mostly lower than 5 mm/month and cumulative displacements 30-100 mm (Figure 6, middle-left). They have experienced some accelerations for the two extreme events, with a displacement rate under 10 mm/month. Low or moderate damage (A or B) is mostly expected.

- Pattern Z: Landslides with a high probability of reactivation and moderate/high intensity, characterized by rates higher than 2-3 mm/month and cumulative displacements greater than 100 mm (Figure 6, lower-left). They have experienced accelerations for the two extreme events, exceeding the displacement rate of 10 mm/month. For the PoR where the road is situated on the crest, we consider that they follow the pattern Z, irrespectively of the inclinometer measurements. High or very damage (C or D) is mostly expected.

Using the intensity-damage correlations of Table 10, the annual probabilities of exceedance of a given damage level were established, as shown in Table 11. Similar patterns were detected too for landslides which are hardly or not all responsive to rainfall events (Figure 6, right), even for the two afore-mentioned extreme events. In that case, higher probability values are set in order to reflect the increased uncertainty for the causes leading to the terrain acceleration (Table 11).




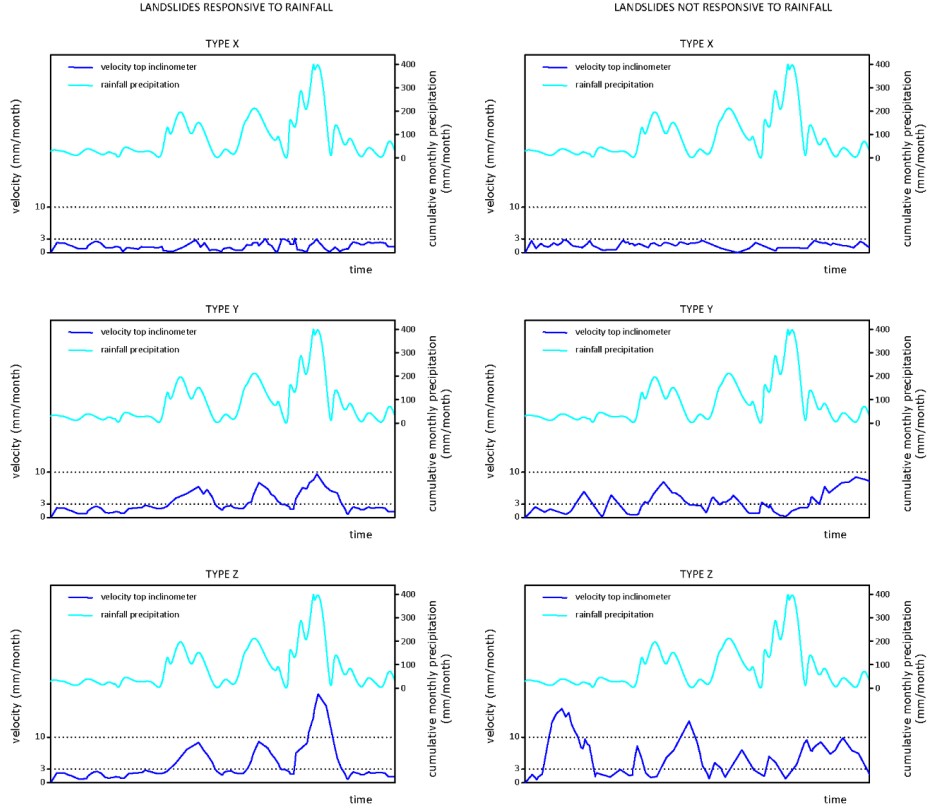

**Figure 6: Patterns of movement for landslides responsive (left) and not responsive (right) to rainfall.**

**Table 11: Annual probability of exceedance of a damage level and (in parenthesis) return period in years.**

|  | Intensity | Type O | Type X | Type Y | Type Z |
|---|---|---|---|---|---|
| Responsive to rainfall | Low ($P_l$) | 0.01 (100) | 1 (1) | 1 (1) | 1 (1) |
|  | Moderate ($P_m$) | 0.01 (100) | 0.02 (50) | 0.5 (2) | 1 (1) |
|  | High ($P_h$) | 0.01 (100) | 0.002 (500) | 0.01 (100) | 0.02 (50) |
|  | Very high ($P_{vh}$) | 0.01 (100) | 0.002 (500) | 0.01 (100) | 0.02 (50) |
| Not responsive to rainfall | Low ($P_l$) | 0.01 (100) | 1 (1) | 1 (1) | 1 (1) |
|  | Moderate ($P_m$) | 0.01 (100) | 0.05 (20) | 1 (1) | 1 (1) |
|  | High ($P_h$) | 0.01 (100) | 0.005 (200) | 0.02 (50) | 0.05 (20) |
|  | Very high ($P_{vh}$) | 0.01 (100) | 0.005 (200) | 0.02 (50) | 0.05 (20) |

The consequences, actions and costs related to slow-moving landslide damage repair that were used for the risk assessment and for the characteristics of the case-study are also reported Table 14 (Appendix). The risk calculation for slow moving landslide is also based on the general Eq. 1. Nevertheless, in this case, for the realistic application of that equation three further points should be taken into consideration:





1) The probabilities of high or very high damage alternate when applying Eq. 1, depending on the location of the road section on the body (high damage) or on the scarp of the landslide (very high damage), and/or the absence (high damage) of presence (very high damage) of a shear crack on the road platform.

2) To calculate the total expected cost, the UC of Table 14 has to be multiplied with the affected road section length (multiples of 10 m), for all damage levels. The affected section length is expected to vary for each damage level, in different percentages of the (total) road length that is marked between the landslide boundaries. For no/slight damage the percentage 10% of the total road length is taken, for moderate damage 20%, for severe damage 50%, and for partial/total destruction 100% (worst case scenario).

The reasons for reducing the affected road section to a percentage of the total road section in the landslide are the following:

i) Some of the landslides in the study area demonstrate composite and complex movements, incorporating multiple smaller sliding bodies, with very local displacements. In that case, instabilities and reactivations take place only locally, affecting only smaller fractions of the road;

ii) The landslide velocity is expected to present variations within the landslide body. More severe damage often demonstrates only at limited sections of the road, corresponding to the local highest movement rates. This might occur mostly for lower velocities, while higher velocities are more probably related with a generalized instability and damage.

iii) Minor displacements can be accommodated by the pavement (especially for flexible pavements), thus they do not result in damage of the entire road section. Major displacements, instead, result in greater stresses and strains for the pavement and more extensive damage, especially for rigid pavements that cannot accommodate them.

3) In the case of slow-moving landslides, the annual probabilities for the different damage levels should be mutually excluding, for the risk calculation. The values of Table 11 provide the annual probability of exceeding a given damage level. Thus the annual probability of damage A, $P_A$, (none/slight) is the annual probability of exceeding damage A minus the probability of exceeding damage B (moderate). Accordingly, the annual probability of damage B, $P_B$, is the annual probability of exceeding damage B minus the probability of exceeding damage C or D (partial/total destruction or severe damage). The annual probability of damage C or D, $P_C$ or $P_D$, is equal to the annual probability of exceeding damage C or D. All probabilities should be in the range [0,1], thus if the result from the above calculations is negative, $P_A$, $P_B$, $P_C$ and $P_D$ are taken as 0.

After incorporating these modifications, Eq. (1) is modified to Eq. (6) for slow moving landslides.

$$R_{SL} = P_l * 0.1 * L * C_l + P_m * 0.5 * L * C_m \quad + P_{h(or\ vh)} * 1.0 * L * C_{h\ (or\ vh)} \quad , \tag{6}$$

Where:

$R_{SL}$: Risk for slow moving landslide

$P_j$: Annual probability of a given damage level (j takes values of l:low, m:moderate, h: high, and vh: very high)

$C_j$: Consequences corresponding to the velocity associated with the given damage level, in terms of UC

L: total length the affected road section (multiples of 10 m).



### 3.4 Failure of sea walls (SW)

The pavement stability is determined by its capacity to resist the maritime storms that act on the coastal front, eroding the rock mass and impacting on protection elements like coastal walls. Four fundamental factors were identified for the stability of the road platform, which are used to define an index called Potential Instability of the Platform, PIP. These are:

(i) The geological structure that affects the slope stability. If the road is built on layers that dip towards the sea cliffs, the erosive action of the wave at the slope toe affects the slope stability and the platform itself. Figure 7 (top-left) presents favourable and unfavourable geological structures, with respect to the orientation of the coastal front. This factor is used to evaluate the existence or absence of a potential failure mechanism, and relates to the cinematic probability of instability.

(ii) The existence or not of a terrain buttress that protects the slope toe from erosion (Figure 7, right). This factor is used to evaluate the effect of resisting forces to instability. The higher it is, the lower the probability of failure is.

(iii) The incidence angle of the waves. The erosive action is expected to be faster when the sea waves are perpendicular to the coast. At the coastline of Gipuzkoa, the common wave directions are W, NW and N (Figure 7, bottom-left). This factor is related to the driving forces inducing instability.

(iv) A correction factor to consider the effect of protection structures against coastal erosion. This factor refers to the increase of resisting forces to instability (Table 12).





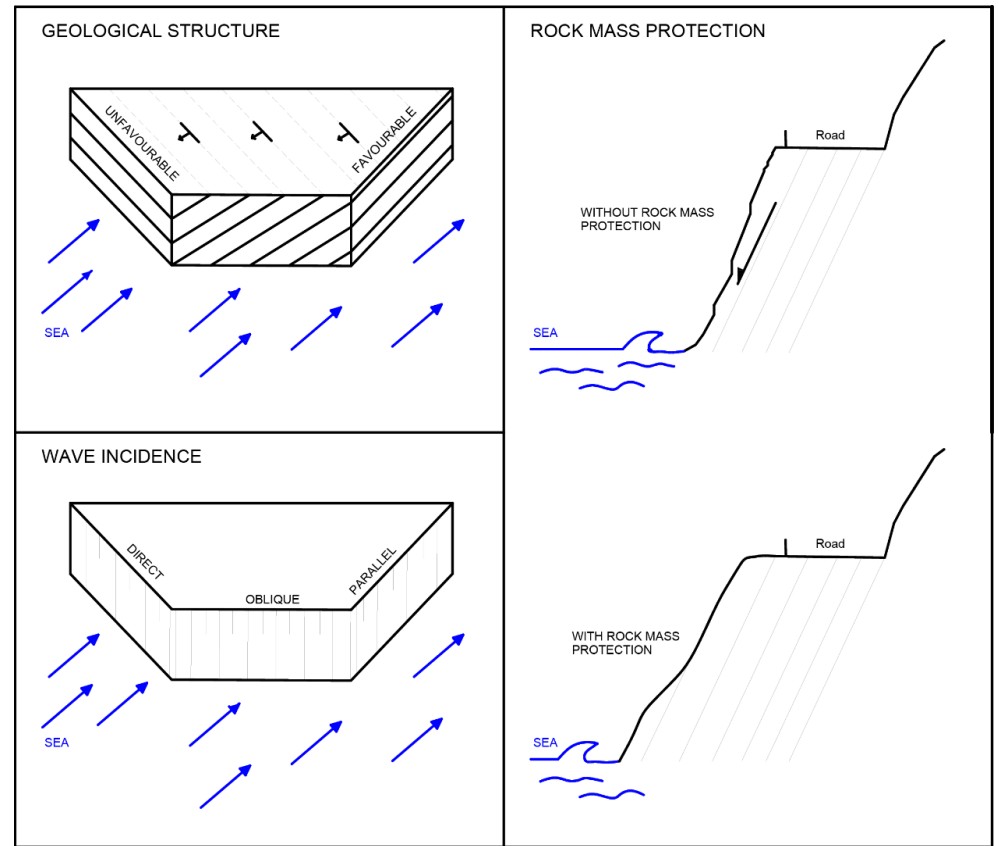

**Figure 7: (top-left) Unfavourable and favourable disposition of the geological structure layers of the slope (direction and orientation), in relation to the orientation of the slope face. Arrows indicate the predominant waves; (bottom-left) Direct, parallel and oblique incidence angle of the waves on the coastal front; (right) Rock mass protection effect without and with a protection mass. In the former case the toe erosion destabilises the coastal slope leading to damage of the road platform.**

The index PIP is first calculated by Eq. (7), with values that range between 0 and 9.

$$PIP = G_s + M_p + I_o + F_c , \tag{7}$$

Where

Geological structure ($G_s$): score from 1 (favourable) to 3 (unfavourable)

Protection mass ($M_p$): score from 1 (sufficient) to 4 (without mass)

Wave incidence ($I_o$): score from 1 (oblique/parallel) to 2 (direct)

Correction factor ($F_c$): values from Table 16.

**Table 12: Correction factor of the PIP.**

| Measure | Score |
|---|---|
| Road embankment wall without anchoring. Coastal front without protection | 0 |





| Road platform with anchorage deeper than the potential rupture surface. Coastal slope front without protection. | -2 |
|---|---|
| Coastal front protected with a masonry wall or reinforced concrete | -2 |
| Coastal front protected with an anchored masonry wall or reinforced concrete | -4 |

The scoring for the factors of Eq. (7) was selected empirically and equal weights were attributed to all factors.

The index PIP was associated with the annual probability of failure sea walls (Table 13) based on expert judgment. Although there is a lack of empirical data to validate this approach, there is certain equivalence between the annual probability of failure for sea walls and retaining walls, which makes its use practical for the comparison of the two hazards. Sea walls with PIP in the range of 0-2 have the same probability of failure as retaining walls with HI lower than 2.6. For constructions of Category III (without specific design that do not follow standards) and IV (poor or completely missing engineering knowledge basis) this corresponds to a safety of factor greater than 1.7. Respectively, for PIP indices 3-6 (corresponding to HI 2.6-3.2), the safety of factor is higher than 1.5, for PIP 7, it is higher than 1.2, and for PIP in the range of 8-9 the safety of factor can be lower than 1.

**Table 13: Annual probability of failure corresponding to the different classes of the PIP.**

| Index PIP | Description | Annual probability of failure |
|---|---|---|
| 0-2 | Low potential of instability. No actions required | $P < 10^{-3}$ |
| 3-6 | Moderate potential of instability. Periodic inspection is required | $10^{-3} < P < 10^{-2}$ |
| 7 | High potential of instability. Further study of the adequacy of the actual protection is required. | $10^{-2} < P < 10^{-1}$ |
| 8-9 | Very high potential of instability. Further protection measures are required. | $P > 10^{-1}$ |

In the proposed approach, the maintenance state of the walls and of the good functioning of the anchorage is not taken into consideration. Still the sea wall failure is often the result of eroded anchorage and bad support. To overcome this limitation a conservative application of the correction scores of Table 12 is recommended.

The consequences of the failure depend on the mobilized volume, which relates to the extent of the affected platform. Table 14 summarizes the actions that are expected for 4 reference volumes. Interpolation and extrapolation is suggested for other volumes. In lack of further data, and as a simplification the length of the affected height can be taken as three times the wall height.

## 4. Application examples and results

The methodology is being applied on a periodic basis to the road network of Gipuzkoa. The calculation of the risk for the different hazards was organised using automatized Excel data sheets, where the user introduced the required data from a closed list of options. Some application examples for selected PoR are presented here. The overall results for all analysed PoR during the 2015 inspection are shown in section 4.5. Due to space limitations, only the most important data needed for the application of the proposed methodology are provided here. An extended archive with further details on the location and type of instabilities, accompanied by images and maps and detailed

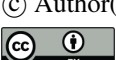



### 4.1 Rockfalls

We indicatively present the risk for two different cases of rockfall affected sections: the PoR L1C, situated at the local road GI-3324 (from km 0.700 to 1.450) and the PoR P4C of the highway N-1 (km 407.395 to 407.680). The two PoR are shown in Figure 8.

Rock instabilities at the PoR L1C are due to the unfavourable structural conditions of the rock mass, leading to

planar and wedge failures, with a safety factor of the blocks on the slope seemingly smaller than 1. In some few areas toppling mechanisms are also observed. Although rockfalls occur with a frequency higher than 2 events per year, there is no historical record of the events. The road platform is in a good state without major impact signs, indicating that no high magnitude events have been occurring. Thus, the magnitude-frequency relation of events here is evaluated considering the slope properties from Table 4, and applying Eq. 2 for the calculation of the $I_F$.

For $I_P$ (joint persistence): Low, $I_{DC}$ (density scars)>0.3, $I_E$ (differential erosion): Yes, $I_{FP}$ (number of points with potential rockfalls)>11, and $I_{SMR}$ (SMR index): 40-80, $I_F$ results to be 6, which corresponds to an annual frequency of events equal to 1 (from Table 5). This frequency is proportionally distributed amongst the magnitude classes A, B and C, according to the in situ observed relative frequency of potential unstable volumes (45% for <A: 0.5m$^3$, 40% for B: 0.5-5 m$^3$, 15% for C: 5-50 m$^3$). This results in annualized expected frequency of events equal to 0.45,

0.40 and 0.15, respectively for each magnitude class. No protection measures exist, thus no correction is made on that frequency. The risk is then calculated for each magnitude size A, B and C and summed up after Eq. (1), considering the UC, of Table 14. The total annual risk for L1C is 1.55.

For the PoR P4C, a similar procedure was followed. For $I_P$: Moderate, $I_{DC}$ >0.16, $I_E$: No, $I_{FP}$: 4, and $I_{SMR}$: 0-40, $I_F$ results equal to 5, which corresponds to an annual frequency of events equal to 0.2. This frequency is distributed

to the magnitude classes: 0.08 for A, 0.09 for B, 0.01 for C, 0.01 for D and 0.01 for E. In this case, a correction is applied on the annual frequency. Considering that the slope is partially protected by gunite and bolts in a bad state of maintenance, and that a ditch with width smaller than 5 m exists, the summative frequency correction factors for the two types of measures from Table 6 are 0.45 and 0.2 for magnitudes A and B. For higher magnitudes, the existing protection measures are not considered to be efficient. The annual frequencies per size after correction are

0.03, 0.07, 0.01, 0.01, 0.01 for A, B, C, D and E, respectively, which after multiplication with the correspondent UC and summing, they result in a higher total annual risk than the previous one, and equal to 3.05. The higher risk is ought to the existence of bigger rock blocks which cannot be retained by the actual protection measures.





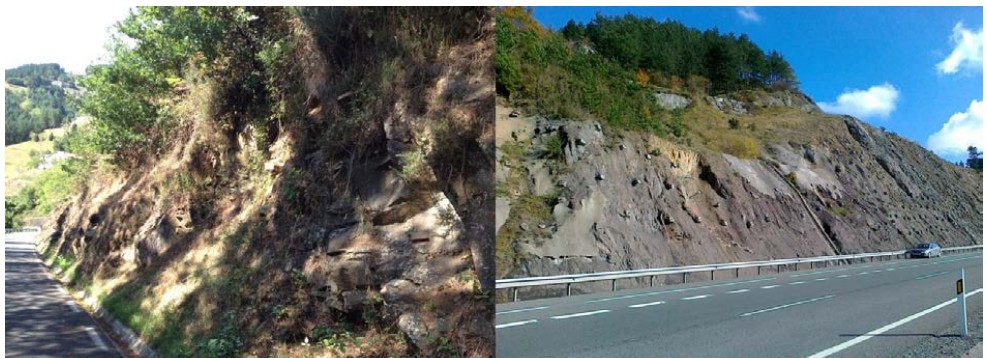

**Figure 8: (left) PoR L1C of the road GI-3324 (from km 0.700 to 1.450), without protection measures; (right) PoR P4C of the road N-1 (km 407.395 to 407.680) with protection measures.**

## 4.2 Anchored retaining structures

Two further examples, for the risk related to the failure of anchored retaining structures are presented (Figure 9).

The first one is the highway section P2A, where two anchored walls sustain a weathered flysch rock mass. The two walls cover the upper and middle part of the slope. The upper wall has a length of 60 m, with 24 anchors and the lower is 42 m long with 42 anchors. The slope has an inclination of 35 degrees in the first 20 m and of 20 degrees in the rest of its length. At the bottom of the slope there is a concrete wall. During the construction works of the N-121-A, water seepage was detected on the middle and upper parts of the slope, which have been related to the mobilization of soil. The slope is monitored with inclinometers, piezometers and the walls with hydraulic load cells, given the existence of an urbanized area on the top of it. In the upper part of the slope there a weathered zone, with an estimated thickness of 20 to 22 m, which determined the type and length of the anchors.

Following the procedure described in section 3.2, for a Safety Factor >2 (1 point from Table 7), working load <55% (1 point), grout length >1.2 m/10 t (1 point), anchoring ground is a mixture of sound and weathered rock (3 points), available data for anchors and technical assistance during construction (1 point), the calculated hazard index from Eq. (5) before correction is 1.222. Given the presence of cracks in the lower wall, a correction factor of 2 is applied to the HI, which becomes 3.222 and the corrected annual probability of failure is then 0.0072 (from Table 8). The upper and lower walls together have a total height of 8 m. When applying the general risk equation the respective costs from Table 14 are considered and adjusted to the total length of the affected road section (24 m or 2.4 x 10 m), and they are found to be 290.4 UC. The total risk for this section, as the product of failure probability with that cost is equal to 2.09.

The second section, the PoR C8A corresponds to a rocky slope of 240 m and height 37m, of an average inclination of 45 degrees. It is treated with gunite with a protection net and a reinforcement of 646 bolts of 16 m length. The works for the reinforcement of this slope were performed between 1992 and 1993, with 8 strips of gunite, 5 m tall each one. Every strip includes 4 rows of bolts, except the lowest one which has 5 rows and the crest which does not have any. The load cells indicate that presently several bolts are overloaded. The rock mass is slightly to

moderately weathered over the entire slope. The maximum volume that has been estimated to be unstable is 21,800 m³.

The parameters that have been used for C8A are (Figure 9, right): Safety Factor <1.30 (it is not known for this case) (5 points), working load >65% of the ultimate load (5 points), grout length >1.2m/10 t  (1 point) and

anchorage on a slight to moderate weathered rock (3 points). Although data exist for some anchors, it is not available for all of them, which is penalized assuming the respective value of "no" in Table 7. Construction was performed with technical assistance. The initially calculated HI is 3.67. As the bolt pressure increase is greater than 65% of the ultimate load (up to 80%), the HI is increased by 2 and it is 5.667 which corresponds to an estimated failure probability of 0.82. This value was calculated according to Table 8, after fitting a power law

curve to the HI and the probability threshold values mentioned therein. As the wall height is H > 6 m and the affected road length is 111 m the total cost is 1343.1 UC and the risk is 1,101.34, which is substantially higher than the previous one, as a consequence of the higher hazard in this section.

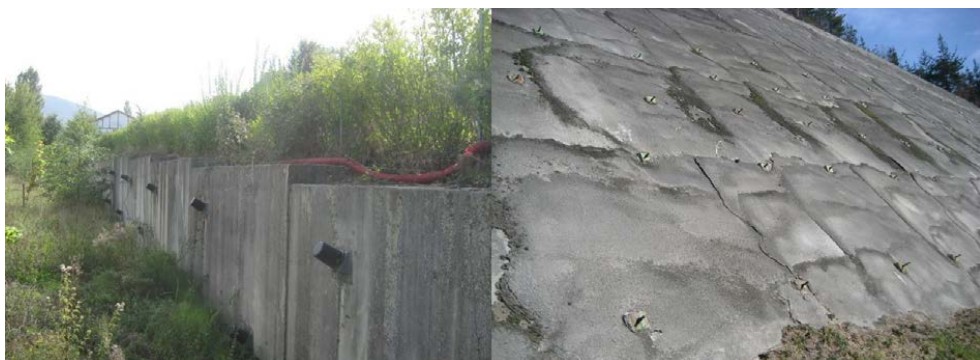

**Figure 9: (left) PoR P2A of the road AP-8-1-1 (from km 77.140 to 77.210) where the crest of the slope and the retaining wall with anchors is seen; (right) C8A of the road G-2634 (32.980 to 33.150) with gunite, rock bolts, and anchors.**

### 4.3 Slow moving landslides

Two examples of PoR, subjected to damage for being situated within slow moving landslide areas are presented here: the C3C and the C9G  (Figure 10).

The PoR C3C involves a rotational landslide, occurring in the interface between clay colluvial soil and rock. It affects a road section of almost 200 m, with a retaining wall of 1 m height, along  41 m. There is not a historical

record of the damage evolution. The road platform is found to be cracked and vertically deformed, as well as the ditch. For about 20 m the cracks are 2-3 cm open and mostly affect one direction of the road. The most probable reason for the instability is the presence of water and the deficient drainage causing humidity and water flows. Slope movements occur during high rainfall episodes, which was also confirmed by the indications of the inclinometers as explained in the following.




There are two inclinometers starting measuring since November 2013, initially with monthly frequency, which are 14.5 and 15.5 m deep each. Movements were found to be concentrated at a depth of 0.5-2.0 m (corresponding to the thickness of the colluvium soil layer). The inclinometer measurements indicate that the soil movements directly respond to the total monthly precipitation intensity, expressed in mm/month, with records of up to 8 mm/month

and cumulative displacement higher than 100 mm. These movement rates correspond to Pattern Z (Figure 6, left). According to Table 11, the annual probability of exceeding low damage is 1, of moderate damage is 1 too, and of high damage is 0.02. As field inspections showed no presence of scarp or shear crack, the potential of very high damage was eliminated. Consequently, the probability of occurrence of only high damage is $P_h$=0.02, of only moderate damage is $P_m$=1-0.02=0.98, and of only low damage is 0 (moderate damage constantly overlaps with

low damage). Applying Eq. (2) and the costs of Table 14 for a total length of affected road of 200 m, the risk result here is 66.18.

The soil instability affecting the C9G is ought as well to colluvial soil accumulations of 3-4 meters depth, over the bedrock. There are visible settlements and the ditches are deformed. The landslide is instrumented with two inclinometers that indicate that it is active.

The inclinometer measurements indicate vertical displacements lower than 2-3 mm and cumulative displacement lower than 30 mm, which points to a movement pattern of type X. There is not any scarp or developed shear displacements. Accordingly the probability of exceeding low damage is 1, moderate damage is 0.05 and high damage is 0.005. As previously, the probability is calculated for high damage: 0.005, for moderate damage: 0.045, and for low damage: 0.5, which for total a length of 300 m, and the same as previously costs, gives a total risk of

10.84.

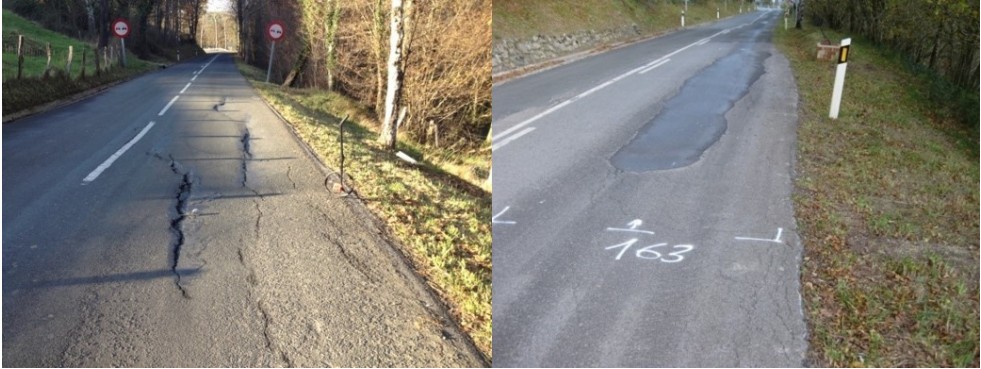

**Figure 10: Road sections situated within slow moving landslides, with damage: (left) PoR C3C of the road GI-2133 (from km 8.280) with observed deformations and cracks on the road pavement; (right) PoR C9G**
**of the road GI2637 (from km 17.891 to 18.220) showing minor cracks already sealed.**

**4.4 Failure of sea walls**

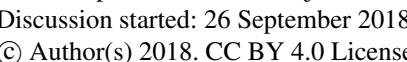



The failure of sea walls and the consequent slope instability has direct effects on the stability of the overlaying road platform at the coastal roads of Gipuzkoa, which is transited by a high number of tourists, especially during the summer months. Here, the PoR B9A and P7L are brought as an example. Figure 11 shows part of those slopes and of the sea walls.

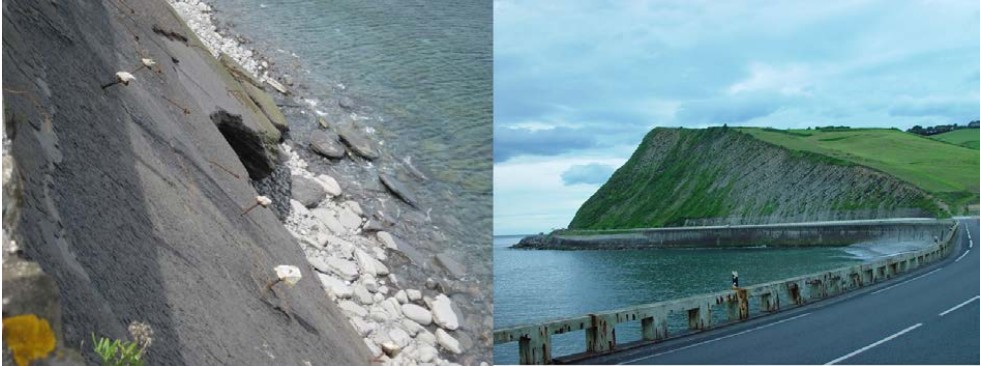

**Figure 11: (left) PoR B9A of the road GI-638 (from km 1.400), of high risk, with a view of the coastal wall from the road, with oxidised and bent bolt. The sea wave action has left the rock exposed and the bolt unsupported; (right) PoR P7L of the road N-634 (from km 24.650 to 24.750), which was stabilized in May**
**2014.**

The section B9A is a 15 m high rocky slope with a 45 degree inclination towards the sea. Sea waves cause foot erosion and loss of support for the overlaying strata. The upper part of the wall is supported by bolts in a very bad maintenance state. The potential failure mechanism is planar failure initiated by the loss of support by erosion of
the stratification. The calculation of the index PIP is done considering the following values for the reference parameters: Geological structure (Gs): 3 (unfavourable), Protection mass (Mp): 4 (without mass), Wave incidence (Io): 2 (direct incidence), and Correction factor (Fc): 0 (Table 12), as in practice the bolts are out of order and there is no protection of the coastal front from the waves. Equation (7) provides a value of 9 for the PIP, which from Table 13 and by extrapolation corresponds to very high potential of instability and annual probability of failure
P=0.1. The cost of the consequences is calculated according to Table 14. An unstable volume of 600 m$^3$ is roughly assumed from topographical measurements, which sets the cost at 159.6 UC per 10 m of affected road length. The potentially affected road is roughly assumed to be 3 times the wall height and equal to 45 m. Thus the expected cost is 718.2, and the risk is 71.82.

At the section P7L and for a specific part of the road, there is a masonry sea wall, 6 to 9 m high for the whole
length of the section (for the calculation we considered 8 m). The exposed slopes have a fair disposition of the joint sets, which dip orthogonal to the sea. As previously an annual probability of failure P=0.01 was calculated for a PIP equal to 6. This PIP is evaluated considering Geological structure (Gs): 2 (fair), lack of protection rock mass (Mp): 4 (without mass), Wave incidence (Io): 2 (direct incidence), and Correction factor (Fc): -2, as the coastal front remains protected by the masonry wall, which is locally reinforced. For the 90 m$^3$ of mobilized mass





corresponding to 34.5 UC per 10 m repair cost, and an affected road length 24 m, the risk for this road is 0.82. The effect of the protection wall in reducing the risk, is reflected by its lower value, when compared to the B9A.

**4.5 Overall results and discussion**

Out of the totally 95 PoR, 20 concern rockfalls, 37 anchored walls, 27 slow moving landslides and 11 sea walls. The classification of the risk here, was based on economic criteria, considering that it expresses the average annual repair cost at a section. Five risk levels were used: low (<1 UC), moderate (1-10 UC), high (10-100 UC), and very high (>100 UC).

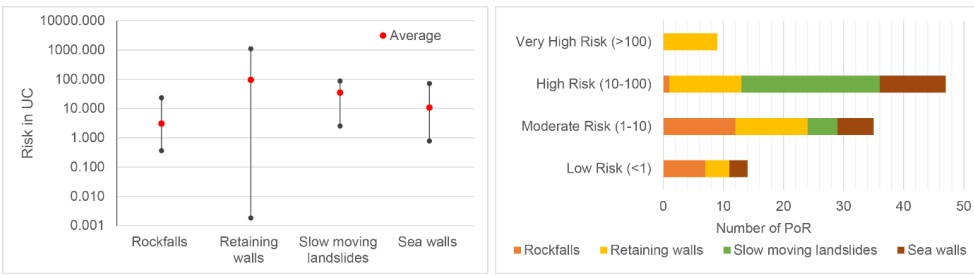

**Figure 12: (left) Average, maximum and minimum calculated risk for each hazard type; (right) Number of PoR per hazard type in each risk class.**

Figure 12 shows how the risk levels are distributed over the different hazard types. High risk areas mostly involve retaining wall failures or slow moving landslides. Rockfalls principally cause low and moderate risk and sea wall failure are of low or high risk. The risk classification highlights 12% of the analysed PoR being of very high risk and needing imminent protection interventions. The highest risks in the study site (above 100 UC) were observed for 9 of the retaining structures. 28% of the PoR fall into the class of high risk, 39% of moderate risk and 21% of

low risk. Depending on the prioritization needs and the economical restrictions for the planning of the interventions, other thresholds can be used for the risk classification, in order to reduce the number of sections marked as of highest priority.

The risks corresponding to the repairs related to the failure of the retaining walls can be one order of magnitude higher than for the rest of the hazards. This is reasonable given the fact that, besides road cleaning and repair costs,

the additional wall construction/repair costs are high and that for this type of events, high soil masses can be mobilized.

The calculated risks overestimate the real average annual costs, as they do not take into consideration that after the hazard is reduced and the chances of damage for the following period are much lower after protection interventions. Moreover, in practice, low and moderate damage are not repaired each time they occur, even if

required, but in larger intervals, which reduces the real repair costs.

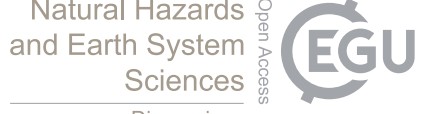

## 5. Conclusions

In this paper a procedure for the quantification of risk related to the geotechnical hazards across a road network has been presented. The studied hazards are rockfalls, failure of retaining structures, slow moving landslides, and failure of sea walls due to sea erosion. The risk has been calculated in monetary terms, as multiples of a Cost of 
5 Unit set at 1000 €

The application of the proposed procedure to the road network of Gipuzkoa highlights that one of the most important aspects of risk quantification is the calculation of the hazard in terms of temporal probability of an event of a given magnitude. In the studied area, the extensive and periodic collection of data permitted the magnitude-frequency evaluation based on historical data and, for rockfalls, where this data lacked, the development and 
10 calibration of an indicator model to assess it. The parameters included in this model are the joint persistence, the density of scars, the differential erosion, the number of points with potential rockfalls, and the slope mass rating SMR index.

For slow moving landslides with permanent or episodic activity, the annual probability of occurrence is not relevant, and it is assumed equal to P=1. Instead the annual probability of reactivation of a landslide with a given 
15 intensity (low, moderate, high, very high) was used for the quantification of the hazard. The landslide velocity was found to correlate well with the visible damage on the road pavement. The monthly thresholds of 3 and 10 mm and the cumulative displacement of 30 and 100 mm were used for the landslide intensity classification (see Table 13). For several PoR of the study area, a good correlation of the soil movements with the daily rainfall intensity was found. This was used as a reference in order to fix the annual probability of reactivation for the landslides, 
20 considering as well their intensity, based on the return period of the extreme rainfall events of 2011 and 2013, which is about 100 years.  The probabilities of occurrence, for each PoR, were then attributed for lower or higher intensity classes, according to the observed landslide pattern with reference to this value, and the maximum monthly velocity and annual displacement observed by the inclinometer measurements. The probability of reactivation for the landslides that were not found responsive to rainfall was considered higher than for the 
25 responsive ones, in order to penalize the uncertainty related to the triggering conditions.

In the study area, the number of sections is very high, not permitting the evaluation of  the probability of slope/structure failure with high precision. Nevertheless the proposed procedure takes into consideration the different failure mechanisms and the factors which are relevant to the probability of failure (rainfall, deformations, and design criteria for the structures). The highest risks in the study area referring to the repair cost for the damage 
30 of roads, are, in most cases associated, in descending order, with retaining structures, slow moving landslides, coastal walls, and rockfalls. The annual repair cost for retaining wall failure presents large variation for the different PoR, ought to the variation on the maintenance conditions and working loads. Using the proposed procedure, the prioritization of interventions for all the PoR was made and the number and location of the PoR that require imminent interventions can be assessed. The thresholds defining the risk classes can be adjusted, 
35 according to financial availability for interventions, so as to point a smaller or higher number of PoR.

The application and calibration of the proposed methodology is an on-going procedure, as inspections are made periodically for the PoR with the higher risk rates, and  landslide activity or damage are being assessed on a




continuous basis, especially after extreme rainfall events. Further work is needed in the following period, to adjust the methodology after the collection of new data.

The calculated risk results are conservative, as in reality low and moderate damage are not repaired each time they occur, but in larger intervals, however the inclusion of this parameter cannot be standardized for the study area, as
the repair works are not regular.

As described in the introduction, the methodology developed in this work had as a starting point the requirements and data availability in the selected case study. Several parts of the proposed procedure for the risk assessment and ranking along road networks are strongly related to local conditions, concerning geological, geomorphological and climatic parameters and are empirical. Accordingly, the thresholds that have been selected here for the hazard
descriptors and the classification of the consequences strongly depend on the expected range of frequency and magnitude/intensity of events in the study area. Moreover, the selection, scoring and weighting of the factors which are used for the calculation of the risk components have an empirical/judgmental basis, and as such, their use, although supported by the physical interpretation of the phenomena, can only become acceptable for the specific case-study on the grounds of calibration and validation. The validation of the proposed classifications and
thresholds is an iterative on-going procedure which requires observing the actual evolution of the phenomena, damage and maintenance/repair costs at the PoR.

In that sense, the application of the proposed methodology to other case studies is principally suggested in terms of procedure and factors to consider for a multi-hazard integrated risk assessment for roadways. Instead adaptation to the local conditions is needed for the scoring, and classification of the hazard parameters, and for the assessment
of temporal probability values considering the intensity and recurrence of local triggering factors, as well as for the asset and cost assessments. Further applications of the procedure presented here to areas with similar or diverse data settings would be useful for its refinement, and would provide an insight for framing the conditions of its transferability to other regions.

**Acknowledgments**

This work has been performed with the support of  project on Integrated Assessment of the Geotechnical Risk of the Gipuzkoa road network (15-ES-563/2009) funded by the Gipuzkoako Foru Aldundia and of the Project RockModels-Caracterización y modelización de los desprendimientos rocosos (https://rockmodels.upc.edu/en) funded by the Spanish Ministry of Economy (Ref. BIA2016-75668-P, AEI/FEDER,UE).

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



**Appendix - Table 14: Consequence classification and costs for each hazard type (source of costs Ikerlur and Gipuzkoako Foru Aldundia).**

| Hazard type and variable for consequence classification | Class | Principal consequences and disruption | Actions | Cost (in UC) |
|---|---|---|---|---|
| RF - Rockfall magnitude (m³) | A (<1) | platform not/partially occupied, no/partial disruption | (Alternative pass + removal debris) | 0.8 |
| | B (1-5) | platform partially occupied, partial disruption | (Alternative pass) + removal debris | 1.5 |
| | C (5-50) | platform occupied, disruption | Alternative pass + removal debris | 3.9 |
| | D (50-500) | platform occupied, disruption | Interruption road + removal debris + slope scaling | 17.5 |
| | E (50-5,000) | platform occupied and damaged, disruption | Interruption road + removal debris + repair road + intensive scaling | 117.2 |
| | F (>5,000) | platform occupied and damaged, disruption | Interruption road + removal debris + repair road + intensive scaling | 172.4 |
| RS - Wall structure failure extent | Partial failure of small wall | no damage, no/partial disruption | Removal + slope scaling | 20.9 (per 10 m wall) |
| | Height wall: ≤ 6 m | platform occupied and damaged, disruption | Removal + slope scaling + stabilization + reconstruction wall + repair road | 70.3 (per 10 m wall) |
| | Height wall: > 6 m | platform occupied and damaged, disruption | Removal + slope scaling stabilization + reconstruction wall + repair road | 121.0 (per 10 m wall) |
| SL - Max rate of terrain displacement (mm/month) and instability indicators | $v_A < 3$ | Without/slight damage, no disruption | Repair road | 1.4 (per 10 m damaged road) |
| | $3 < v_B < 10$ | Moderate damage, no disruption | Repair road | 15.0 (per 10 m damaged road) |
| | $v_C > 10$ | Severe damage, no/partial disruption | Repair road + stabilization | 37.4 (per 10 m damaged road) |
| | $v_D > 10$ and presence of scarp or developed shear displacements | Partial/total destruction, disruption | Interruption road + scaling slope + stabilization | 45.4 (per 10 m damaged road) |
| SW - Mobilized volume (m³) | 40 | Without erosion, no disruption | Interruption road + scaling slope | 17.9 |
| | 90 | Without erosion, disruption | Interruption road + intensive scaling slope | 34.5 |
| | 400 | Erosion, disruption | Interruption road + stabilization | 82.7 |
| | 600 | Erosion, road damage, disruption | Interruption road + stabilization | 159.6 |