# Peer review of "Integrated risk assessment due to slope instabilities at the roadway network of Gipuzkoa, Basque Country"

_Natural Hazards and Earth System Sciences, 2018_

## Referee Comment (RC1) · Anonymous Referee #1 · 16 Oct 2018

The objective of this manuscript concerns the risk assessment characterizing a roadway network affected by different hazards in Spain. For this, a "hybrid" quali-quantitative method mainly expressed in monetary terms has been proposed (as a multiple of a cost unit amounting to 1,000 EUR). The failure probability is evaluated with reference to four types of geotechnical hazards (rockfalls, anchored wall failures, slow landslides, and sea wall failures), whereas the related risk level is based on an economic criterion expressed as the average annual repair cost, at the considered section. The annual final risk is given by the product of probability/frequency occurrences of events, in assigned magnitude classes, per the monetary value of the elements at risk.

[Figure]

Even though the work is interesting, there are some questions which I would like to highlight.

I don't understand why the Authors put together the rockfalls (natural hazard) with the failures of man-made structures such as anchored walls, road platforms and sea walls. It is clear that these failures can be caused by natural causes linked to the slope movements but, in such a case, I would have expected the analysis of failures of passive retaining structures (barriers, fences, etc.), due to rockfalls.

Another big concern is for using heuristic methods in order to calculate the annual P/F occurrence for addressing the lack of adequate data. It seems to me that sometimes they are based on not adequately justified or tested expert judgments. This is particularly true for rockfalls and slow landslides types, where these approaches are used in order to overcome the lack of complete rockfall and landslide catalogues. In this respect, the Authors extend the frequency/magnitude relationships, calculated for sites with adequate data, to slopes with similar geostructural characteristics, scars, heights and block sizes (see on page 9, line 14).

With reference to rockfalls, the scoring assignment for the frequency index (IF) calculation involves five frequency indicators both qualitatively and quantitatively defined. In my opinion, this is a hybrid approach linking qualitative terms to quantitative data. With reference to the Differential Erosion indicator please, clarify what is the used figure for the score 2 (No, Yes, and...?) Then, the scores are summed up to calculate IF (see eq. 2), and a relationship between IF and the rockfall frequency, for enabling the assessment of thresholds, has been established. For this purpose, a calibration has been performed. I thought it would be interesting if the Authors would speak to us the calibration procedure and results.

Also with reference to the correction factors (Fr) assigned to different protection measures (see Table 6 – please, check the correct numbering of all tables and figures; the figure 4 is several times duplicated!), it seems to me that this was done to ensure too

Interactive
comment

high safety standards. I understand that, lacking in literature adequate tested values, the Authors adopted precautionary data but (e.g. see the tunnel case) assuming a range of values between 4 and 2, for magnitude classes A-D it seems to me too high!

Similar considerations must be done for the failure of retaining structures, where the HI factor is only based on subjective evaluations and expert judgments. Hybrid data (qualitative and quantitative in nature) is also reported in Table 7. What means sound rock or mixture? For instance, is it adequate the use of the Schmidt hammer or pocket penetrometer for their characterization? As the Authors are a well-known team working in the field of the hazard and risk assessment, I think that they must be very wary of suggesting not well-tested approaches because they have big authority in this very poorly explored research field.

Concerning the slow movements affecting the roads, might the slight/moderate damage on the road be due to shallow subsidence of the subgrade? (as it appears to me by some photos). Since the inclinometers are very often affected by installation problem, or malfunctioning causing lack of data have you been performing a reliability analysis?

For the failure of sea walls, I think that in an oversimplified way the procedure for the PIP index calculation is evaluated. As it is well know, the undercutting by waves is very important in causing the sea-cliff retreat or wall failure (mainly for toppling). Waves erode the cliff toe, undercutting and over steepening it. This destabilizes the overlying slope, causing it to collapse. Also with reference to the sea walls, the main failure mechanism is linked to erosion by waves. Consequently, the main factors affecting this failure process are the real dynamic pressure exerted by the water at the wall toe, the mechanical strength of concretes and design characteristics. As these quantities are very difficult to assess, generally the research approach uses aerial or satellite photos, topographic survey comparisons, LIDAR techniques, etc. Also the on-shore wave characteristics and meteorological observations in time and space are needed. With reference to eq. 7, and according to my opinion, the protection mass index (Mp) already should incorporate the correction factor (Fc) for the protection structure. What

do you think? I suggest that you remove this hazard from the text.

In conclusion, the suggested approach must be considered as a first attempt that cannot be extended to areas with different geotechnical and geomechanical characteristics, respect to the studied ones. The study confirms that a reliable quantitative risk analysis involving man-made structures can be performed using reliable and numerous data only. Otherwise and for wide areas, only heuristic approaches based on expert judgments can be used. But the question we have to ask is whether it is worth using complex procedures which incorporate not yet well-tested ratings.

—————————————————————

---

## Referee Comment (RC2) · Anonymous Referee #2 · 22 Oct 2018

Review of the manuscript Integrated risk assessment due to slope instabilities at the roadway network of Gipuzkoa, Basque Country

By Olga Mavrouli and co-authors

The manuscript presents a quantitative / qualitative procedure to assess hazard and vulnerability and their integration into risk calculations for roads by considering four different hazardous processes: rockfalls, retaining walls, slow moving landslides, and coastal erosion induced failures.

The work is well written and the subject of the manuscript is of interest for NHESS, however, the manuscript needs some moderate revisions before to be accepted to be

published.

Major comments

The manuscript is basically focused on the hazard evaluation, using quantitative and mainly heuristic approaches. The cost of direct road and retaining structures are not explored along the work, as they were obtained directly from the regional administration.

The few works worldwide dealing with both direct and indirect costs resulting from road damage by landslides have shown that indirect costs can be orders of magnitude higher than direct costs. Although authors clearly state they do not address indirect costs, this topic should be highlighted in the discussion section of the paper.

The methodological section of the manuscript is well balanced, but section 4 is not very well balanced. Section 4.5 (overall results and discussion) is poorly developed when compared with the description of the 8 studied cases for which the risk was calculated. The discussion can be enlarged and improved namely by incorporating the sensitivity analysis of the heuristic options used to assess the hazard.

The conclusion section also need to be improved. The first six paragraphs are not conclusions but a summary of the work.

Minor comments

References are missing along the description of the geology of the study area (Page 6. Line 9 – 18).

Page 6 – line 6-8 The text is not clear. Explain better the relationship between lithology and landforms in the study area.

In section 2, authors should provide the number of Points of Risk corresponding to each considered hazardous process (rockfalls, retaining walls, slow moving landslides, and coastal erosion induced failures). The relevant information is provided in the

manuscript only in section 4.4 (page 29).

What is the difference between hazards and instability mechanisms that are referred in caption of table 2?

Table 2 is not very much informative. Authors can provide the number of PoR considered for each instability mechanism class considered.

In the first part of section 3 - General Methodology for the Risk Assessment, authors should state that different procedures are used to assess magnitude and frequency for the processes that are presented in the next sections.

Figure 3 is nor referred in the text.

Table 3 – How relates the events indicated in table 3 with the 95 PoR referred before? Are some of them the same? It is not clear.

Page 10, line 1-4 The typical size of slope face is uniform? To each extent a scale effect may control the extent of observed discontinuities in the rock mass?

In table 5 The maximum IF should be fixed as 9?

Page 11, line 1-3, Authors state: "To assess the expected frequency ðİŘźðİŘźðİŚ§ðİŚ§ of a given rock[fall] size, the total expected number of events is distributed over the rockfall volume classes, as observed in situ. If this datum is lacking, the proportion of a modal size of blocks multiplied by the total annual frequency is suggested instead. The statement is not enough clear. Please, provide more information regarding this topic.

Page 12, line 2 "The six magnitude classes", instead of "the five magnitude classes".

Apparently, some data is missing in table 7. It is not clear which score applies to 'yes' / 'no' for the project and construction. Therefore, it is difficult to solve equation (5).

Page 17 , line 17-23 Rainfall is not continuous in time neither in space. Please, provide where the rainfall data was registered. Also, authors state that the rainy period of

January-February 2013 that was characterized by moderate to low daily intensity but of long duration, with cumulative precipitation measurements that exceeded the maximums of the reference period 1971-2000. How many weeks refer the long duration?

It is not clear the way authors obtained the intensity classes referred in table 11.

Page 22, line 15 Table 12 instead of Table 16.

Concerning the sea wall failures, is there any relation between the annual probability of failures and the magnitude (size) of the failures?

---

## Referee Comment (RC3) · Anonymous Referee #3 · 5 Nov 2018

Journal: NHESS

Title: Integrated risk assessment due to slope instabilities at the roadway network of Gipuzkoa, Basque Country

Author(s): Olga Mavrouli et al.

General comments

The paper presents a model of evaluation of multirisk in selected points of the route network of the roadway network of Gipuzkoa, Basque Country. The model tries to quantify risks in four scenarios pertaining to about 100 already detected point of risk. The trans-

formation of the inhomogeneous and scarce data in a quantitative score is made up with criteria driven by the judgement and on the basis of the historical available data about instability on the roadway network. So the model is a heuristic one, currently not yet validate. The paper is well articulated and developed, but generates some perplexity. Specific comments First of all, the referee #3 agree with all the specific comments on the submitted paper from referees #1 and 2. One main concern is about the use of the velocity concept in several part of the text and particularly in the section about the low landslides. Although velocity is widely adopted in many landslide classification systems, it is well known (from Physics) that in a force system the velocity of a rigid body mass is not representative of its equilibrium or disequilibrium state, which is demanded to the first derivative or gradient of the velocity vs time. About the classification and risk management of the "slow landslides", the main references to this kind of landslide are the monthly slope deformation (periodically measured in inclinometers) and the one year cumulate displacements. So the Authors indirectly and correctly use the terms that contribute to the velocity vs time gradient (the acceleration). Some points are not clear or instill some doubts: We start with the consideration that number and position of the available instruments in each site are well positioned and representative of the deformative field of the whole landslide body; are the deformation readings in the inclinometers constantly and regularly (which is the frequency) performed? Do the deformations of the inclinometer allow durability of the measures? Bigger movements are usually at the head, but do we analyze the correct one phenomenology, i.e. a shallow instead of a most dangerous and deeper incoming failure? Finally: measurements normally refer to a pre-peak failure stage: the post peak behaviour with the typical range of strength reduction (and then the hazard magnitude) is strongly controlled by the dominant lithology, which has been missed throughout the decisional points of the whole proposed model. Lithology and soil plasticity are common factors in many heuristic hazard models. Similar consideration should be extended to short term vs long term groundwater variations.

The model has been applied to a system spatially extended for about 10.000 sq kilometres and developing several hundreds of kilometres (see fig.1), whose health and wellness rely on a limited number of instrumentations and on periodical inspections; it is characterized by several geological frames and by a well-developed surface hydrography. The local control apparatus is aged about 16 years and does not allow real time measurements in spite of its expensiveness. In other words, the monitoring system installed and which is the main source of data on which the proposed model works can be considered somewhat obsolete. Today the large areas can be controlled by means of active or passive PS INSAR techniques (measurement return time about 6 days); inclinometers can be flanked by optical or capacitive TDR and other kind of devices able of continuous and real-time response.

While the meticulous and complete care in the model deserves great appreciation, some perplexities already expressed, together with the considerations about the repeatability of the scenarios, significantly limit the usefulness of the model in future scenarios.

Technical corrections

In fig.6, Patterns of movement for landslides responsive and not responsive to rainfall, nowhere is reported a numerical scale (also indicative) of time.

---

## Author Comment (AC1) · 12 Nov 2018

Q1. The objective of this manuscript concerns the risk assessment characterizing a roadway network affected by different hazards in Spain. For this, a "hybrid" quali-quantitative method mainly expressed in monetary terms has been proposed (as a multiple of a cost unit amounting to 1,000 EUR). The failure probability is evaluated with reference to four types of geotechnical hazards (rockfalls, anchored wall failures, slow landslides, and sea wall failures), whereas the related risk level is based on an economic criterion expressed as the average annual repair cost, at the considered section. The annual final risk is given by the product of probability/frequency oc-

currences of events, in assigned magnitude classes, per the monetary value of the elements at risk. Even though the work is interesting, there are some questions which I would like to highlight.

A1.We would like to thank the Reviewer 1 for his/her comments and insights, which helped us a lot to improve the quality of the manuscript. We tried to address all comments and suggestions. Please find below the point-to-point response.

Q2. I don't understand why the Authors put together the rockfalls (natural hazard) with the failures of man-made structures such as anchored walls, road platforms and sea walls. It is clear that these failures can be caused by natural causes linked to the slope movements but, in such a case, I would have expected the analysis of failures of passive retaining structures (barriers, fences, etc.), due to rockfalls.

A2.The hazards that were addressed at this work are the hazards that are mostly encountered in the road network of the study area, and are related to the major intervention costs. The challenge in this case, and the motivation for this work was indeed to analyse all hazards at a common scale, so as to prioritize those road sections that need urgent interventions against other where the risk is lower. Putting together all these hazards and comparing the related risk has been the challenge due to the fact that the hazards generated by these processes follow different temporal patterns and intensities have different type and extend of consequences. To the author's knowledge this is the first time that risk from different hazards is calculated in a quantitative way. This is extensively explained in the introduction.

Q3.Another big concern is for using heuristic methods in order to calculate the annual P/F occurrence for addressing the lack of adequate data. It seems to me that sometimes they are based on not adequately justified or tested expert judgments. This is particularly true for rockfalls and slow landslides types, where these approaches are used in order to overcome the lack of complete rockfall and landslide catalogues. In this respect, the Authors extend the frequency/magnitude relationships, calculated for

sites with adequate data, to slopes with similar geostructural characteristics, scars, heights and block sizes (see on page 9, line 14).

A3.The approaches followed, except for the risk related to sea walls, is not heuristic, instead it is based on analogs and proportionalities for calculating the probability of occurrence of the events of a given magnitude/intensity which is assessed based on quantitative data. This procedure is not heuristic. In any case, quantitative analysis can be based on expert judgment as well. Most of the procedures for risk assessment are based on analogs. As for example, the most well-known approach, which is the Rockfall Hazard Rating System follows the same logic, as it assigns the same level of risk to slopes showing similar geometric and geomechanical features. In our work, for the rockfall hazard, we followed such a logic, starting from the slopes with F-M available and assigning these values to slopes showing similar characteristics. This assumption is not restrictive but justified on the grounds of geomechanical characterization and its relation to the hazard, as for the RHRS mentioned above. A similar approach is followed for the slow-moving landslide, where we try to identify their pattern movements and their relation to the triggering factor (rainfall in this case), the latter with a known recurrence. This procedure is in the same line with the calculation of the occurrence of landslides, based on the recurrence of the triggering rainfalls, which is widely used.

Q4. With reference to rockfalls, the scoring assignment for the frequency index (IF) calculation involves five frequency indicators both qualitatively and quantitatively defined. In my opinion, this is a hybrid approach linking qualitative terms to quantitative data. With reference to the Differential Erosion indicator please, clarify what is the used figure for the score 2 (No, Yes, and...?). Then, the scores are summed up to calculate IF (see eq. 2), and a relationship between IF and the rockfall frequency, for enabling the assessment of thresholds, has been established. For this purpose, a calibration has been performed. I thought it would be interesting if the Authors would speak to us the calibration procedure and results.

A4.The index IF is not a frequency indicator but an indicator corresponding to the geomechanical characterization of the slope (please see also previous answers), thus is used to provide 5 different geomechanical characterizations, which after that, they are assigned with a frequency, considering that slopes in the same geomechanical class have common frequency and equal to the one available in some of the sections. Frequency is obtained quantitatively and not qualitatively. The authors recognize that in the way that the text was written, the term "calibration" is not representative of the procedure explained here, which is the followed procedure. For this we rephrased it in the text. On the contrary of what the reviewer mentions here, we do not link qualitative terms to quantitative data, but vice versa, we link quantitative data (frequency) to already defined qualitative descriptors (geomechanical class). We reference to the differential erosion, if it is present the score for the IF is 1 and if there is not it is 0, which means that it does not add to the IF. It is not the scores 0, 1,or 2 which are assigned to the indeces, but the indices get qualities that correspond to a score. In this sense the qualities of the index IE are no or yes, which correspond to the scores 0 or 1, and 2 is not relevant here, that is why there is not quality in the relevant. We added "not applicable" to avoid confusion.

Q5. Also with reference to the correction factors (Fr) assigned to different protection measures, it seems to me that this was done to ensure too high safety standards (See Table 6). I understand that, lacking in literature adequate tested values, the Authors adopted precautionary data but (e.g. see the tunnel case) assuming a range of values between 4 and 2, for magnitude classes A-D it seems to me too high!

A5. As also mentioned in our previous response, the correction factors were assigned to different protection measures based on the observed frequencies in the study area, and in particular at the sections that this data was available. Volkwein et al. (2011), report capacity for energy absoprtion about 3000 kJ, without cushion materials, and about 5000 with it. These capacities are sufficient to retain most volumes of the categories A and B, considering the potential energy of the rock, for falling heights of few tens of meters, as the heights of the rocky slopes in the area,. For magnitudes

C and D, given the fragmentation of the initial rock mass a percentage of the blocks is expected to be retained. Reference: Volkwein, A., Schellenberg, K., Labiouse, V., Agliardi, F., Berger, F., Bourrier, F., ... & Jaboyedoff, M. (2011). Rockfall characterisation and structural protection-a review. Natural Hazards and Earth System Sciences, 11, p-2617.

Q6. Similar considerations must be done for the failure of retaining structures, where the HI factor is only based on subjective evaluations and expert judgments. Hybrid data (qualitative and quantitative in nature) is also reported in Table 7. What means sound rock or mixture? For instance, is it adequate the use of the Schmidt hammer or pocket penetrometer for their characterization? As the Authors area well-known team working in the field of the hazard and risk assessment, I think that they must be very wary of suggesting not well-tested approaches because they have big authority in this very poorly explored research field.

A6. For assessing the probability of failure of retaining structures, we adopted the approach proposed by Silva et al. 2008, who argues that the probability of failure of structures can be assessed, considering besides the Safety of Factor, the quality of the construction. Similarly the HI factor is calculated based on an objective indicator model, including data from the load cells, and the specifications of the technical project. The approach is not based on expert judgement, but clear criteria. Sound rock or mixture refers to the ground where the structure is anchored, and is used as a factor to measure effectiveness and uncertainties for the support of the anchors. Schmidt hammer is not used. We are based on the geotechnical reports of the works.

Q7. Concerning the slow movements affecting the roads, might the slight/moderate damage on the road be due to shallow subsidence of the subgrade? (as it appears to me by some photos). Since the inclinometers are very often affected by installation problem, or malfunctioning causing lack of data have you been performing a reliability analysis?

A7. We would like to clarify that in all the cases analysed here, the road traverses well identified landslides, as such movements are associated to landslide activity. This part was added to the text. The methodology that we propose here for the evaluation of the probability of landslide reactivation based on the indications of the inclinometers can be applied given that the inclinometers function well and are reliable. Indeed, in the study area there are some inclinometers which are inoperative or not reliable. However, the analysis presented here in this paper, refers to those sections with good inclinometer data. Please not, that according to yours and other Reviewers's suggestion, we added up some text in the Discussion section, on the reliability of the inclinometers.

Q8. For the failure of sea walls, I think that in an oversimplified way the procedure for the PIP index calculation is evaluated. As it is well know, the undercutting by waves is very important in causing the sea-cliff retreat or wall failure (mainly for toppling). Waves erode the cliff toe, undercutting and over steepening it. This destabilizes the overlying slope, causing it to collapse. Also with reference to the sea walls, the main failure mechanism is linked to erosion by waves. Consequently, the main factors affecting this failure process are the real dynamic pressure exerted by the water at the wall toe, the mechanical strength of concretes and design characteristics. As these quantities are very difficult to assess, generally the research approach uses aerial or satellite photos, topographic survey comparisons, LIDAR techniques, etc. Also the on-shore wave characteristics and meteorological observations in time and space are needed. With reference to eq. 7, and according to my opinion, the protection mass index (Mp) already should incorporate the correction factor (Fc) for the protection structure. What do you think? I suggest that you remove this hazard from the text.

A8. The reviewer is right, and we also on removing this hazard from the new version of the manuscript.

Q9. In conclusion, the suggested approach must be considered as a first attempt that cannot be extended to areas with different geotechnical and geomechanical character-istics, respect to the studied ones. The study confirms that a reliable quantitative risk

analysis involving man-made structures can be performed using reliable and numerous data only. Otherwise and for wide areas, only heuristic approaches based on expert judgments can be used. But the question we have to ask is whether it is worth using complex procedures which incorporate not yet well-tested ratings.

A9. The authors would like to thank again the reviewer for the comments. For the reasons exposed in the introduction we think that quantitative risk analysis for roads, involving manmade structures is of wide interest. To the authors' knowledge, this is the first time that risk from different hazards is calculated in a quantitative way. Of course, several challenges exist which are also exposed in the introduction. It has to be mentioned that in the study area, there is an extensive and systematic collection of data, which has allowed the development of the proposed procedures, not always available in other study areas. Given the existence of this data, we think that the development of the approaches presented here, even if they leave a margin for refinement and validation is the added value if this work, permitting objectivity and reproducibility in the assessment. However, to avoid any misunderstanding we stress again in the conclusions that the procedures were development here given the local characteristics and available data in the area, and that direct transfer of the results to other study areas, without prior study of the local characteristics, is not recommended.

Q10. (see Table 6 – please, check the correct numbering of all tables and figures; the figure 4 is several times duplicated!)

A10.The numbering of Tables and Figures was checked and adapted to the changes in the text. In our version, we do not see Figure 4 duplicated.

Please also note the supplement to this comment:
https://www.nat-hazards-earth-syst-sci-discuss.net/nhess-2018-234/nhess-2018-234-AC1-supplement.pdf

———————————————

2018-234, 2018.

---

## Author Comment (AC2) · 12 Nov 2018

Q1. The manuscript presents a quantitative / qualitative procedure to assess hazard and vulnerability and their integration into risk calculations for roads by considering four different hazardous processes: rockfalls, retaining walls, slow moving landslides, and coastal erosion induced failures. The work is well written and the subject of the manuscript is of interest for NHESS, however, the manuscript needs some moderate revisions before to be accepted to be published.

A1. We would like to thank the Reviewer 2 for his/her comments and insights, which helped us a lot to improve the quality of the manuscript. We tried to take all the sug-

gestions of the Reviewer into account and prepared a new version integrating them.

Q2. Major comments: The manuscript is basically focused on the hazard evaluation, using quantitative and mainly heuristic approaches. The cost of direct road and retaining structures are not explored along the work, as they were obtained directly from the regional administration.

A2. The approaches followed, except for the risk related to sea walls is not heuristic, instead it is based on analogs and proportionalities for calculating the probability of occurrence of the events of a given magnitude/intensity which is assessed based on quantitative data. Most of the procedures for risk assessment are based on analogs. As for example, the most well-known approach, which is the Rockfall Hazard Rating System follows the same logic, as it assigns the same level of risk to slopes showing similar geometric and geomechanical features. In our work, for the rockfall hazard, we followed such a logic, starting from the slopes with F-M available and assigning these values to slopes showing similar characteristics. This assumption is not restrictive but justified on the grounds of geomechanical characterization and its relation to the hazard, as for the RHRS mentioned above. What we would like to stress here, is that the criteria established for the quantitative risk assessment proposed here are transparent, reproducible, and they are based on quantitative data. The cost of repair for roads and retaining structures was obtained by the co-authors engineering companies and regional administration, based on real repair costs, by previous interventions. Its calculation is based on the works that have to be made for each types consequences. These works are explored and detailed in the Table of the Annex. We think that it is no use for the readers to the actual costs in monetary terms here, as these prices fluctuate a lot, according to location and time.

Q3. The few works worldwide dealing with both direct and indirect costs resulting from road damage by landslides have shown that indirect costs can be orders of magnitude higher than direct costs. Although authors clearly state they do not address indirect costs, this topic should be highlighted in the discussion section of the paper.

A3. We agree with the Reviewer. Indeed indirect costs are very significant when calculating economical loss due to landslide road disruption. We highlighted this section in the discussion section as suggested by the Reviewer.

Q4. The methodological section of the manuscript is well balanced, but section 4 is not very well balanced. Section 4.5 (overall results and discussion) is poorly developed when compared with the description of the 8 studied cases for which the risk was calculated. The discussion can be enlarged and improved namely by incorporating the sensitivity analysis of the heuristic options used to assess the hazard. The conclusion section also need to be improved. The first six paragraphs are not conclusions but a summary of the work.

A4. We enriched the discussion section, with comments related to the uncertainties which are included in the proposed procedure. A sensitivity analysis, showing in quantitative terms the effect of the uncertainties in the estimation of the risk components and the final risk would require systematic information on the data quality, which we do not have available. For this reason, and according to the Reviewers′ most useful comments, we included some comments on the most important uncertainties involved in the analysis. As for the conclusion section, we tried to eliminate information which is repeated in the text, however we have left the most important results from the application.

Q5. Minor comments References are missing along the description of the geology of the study area (Page 6. Line 9 – 18).

A5. The references for the geology of the study area are provided in page 5.

Q6. Page 6 – line 6-8 The text is not clear. Explain better the relationship between lithology and landforms in the study area.

A6. It was rephrased.

Q7. In section 2, authors should provide the number of Points of Risk corresponding

to each considered hazardous process (rockfalls, retaining walls, slow moving land-slides, and coastal erosion induced failures). The relevant information is provided in the manuscript only in section 4.4 (page 29).

A7. Done

Q8. What is the difference between hazards and instability mechanisms that are referred in caption of table 2? Table 2 is not very much informative. Authors can provide the number of PoR considered for each instability mechanism class considered.

A8. The reviewer is right. As part of this information is repeated in the Appendix, we deleted this table.

Q9. In the first part of section 3 - General Methodology for the Risk Assessment, authors should state that different procedures are used to assess magnitude and frequency for the processes that are presented in the next sections.

A9. Done

Q10. Figure 3 is nor referred in the text.

A10. We added reference for Figure 3 in the text.

Q11. Table 3 – How relates the events indicated in table 3 with the 95 PoR referred before? Are some of them the same? It is not clear.

A11. Please note, that as we deleted the part referring to the failure of sea walls, we are now referring to 84 PoR instead of 95 in our manuscript. Out of them 20 concern rockfalls. These latter are distributed in the entire network and not only in the sections of the road N-634. The events indicated in Table 3, instead refer only to the 5 PoR of the road N-634. This was clarified in the text too.

Q12. Page 10, line 1-4 The typical size of slope face is uniform? To each extent a scale effect may control the extent of observed discontinuities in the rock mass?

A12. The slopes have in principle an altitude of few tens of meters and large lateral extension, up to tens and hundreds of meters. As a result, there are no persistence limitations due to topography.

Q13. In table 5 The maximum IF should be fixed as 9?

A13. Indeed. This was also indicated in Table 9, in the revised version

Q14. Page11,line1-3,Authors state: "To assess the expected frequency of a given rock[fall] size, the total expected number of events is distributed over the rockfall volume classes, as observed in situ. If this datum is lacking, the proportion of a modal size of blocks multiplied by the total annual frequency is suggested instead. The statement is not enough clear. Please, provide more information regarding this topic.

A14. This part was rephrased to: During the in situ inspections, data is collected for the relative frequency (%) of potential rockfalls, per volume class and for the most frequently encountered rockfall size on the slope (modal). The annual number of events per volume class is calculated by multiplying the total annual frequency with the relative frequency of each class. If the relative frequency data has not been collected, risk calculations are made, for the modal size, as an average approximation.

Q15. Page 12, line 2 "The six magnitude classes", instead of "the five magnitude classes". Apparently, some data is missing in table 7. It is not clear which score applies to 'yes' / 'no' for the project and construction. Therefore, it is difficult to solve equation (5).

A15. The Reviewer is right. "Five" was replaced by "six", as it is six classes. There are three possible scores for PQ. 1 if "available data for anchors" is yes and "Technical assistance during construction" is yes, 3: if "available data for anchors" is no and "Technical assistance during construction" is yes, and 5: if available data for anchors" is no and "Technical assistance during construction" is no.

Q16. Page 17 , line 17-23 Rainfall is not continuous in time neither in space. Please,

provide where the rainfall data was registered. Also, authors state that the rainy period of January-February 2013 that was characterized by moderate to low daily intensity but of long duration, with cumulative precipitation measurements that exceeded the maximums of the reference period 1971-2000. How many weeks refer the long duration?

A16. Rainfall data that is mentioned here was collected at the Añarbe Dam (inside the study area). Unfortunately, we do not have available official data for the exact number of rain days in this period, as it also depends on the location inside the study area.

Q17. It is not clear the way authors obtained the intensity classes referred in table 11.

A17. The four intensity classes were established so as to correspond to the four damage classes in Figure 5, which in turn correspond to different repair costs. Damage levels were related with the intensity classes, based on observations of the damage on the road and the inclinometer indications.

Q18. Page 22, line 15 Table 12 instead of Table 16. Concerning the sea wall failures, is there any relation between the annual probability of failures and the magnitude (size) of the failures?

A18. After the suggestions of Reviewer 1, this section was removed. In any case, to provide an answer to Reviewer 2, there is no relation between the annual probability of failure and their size.

Please also note the supplement to this comment:
https://www.nat-hazards-earth-syst-sci-discuss.net/nhess-2018-234/nhess-2018-234-AC2-supplement.pdf

---

## Author Comment (AC3) · 12 Nov 2018

Q1. The paper presents a model of evaluation of multirisk in selected points of the route network of the roadway network of Gipuzkoa, Basque Country. The model tries to quantify risks in four scenarios pertaining to about 100 already detected point of risk. The tran formation of the inhomogeneous and scarce data in a quantitative score is made up with criteria driven by the judgement and on the basis of the historical available data about instability on the roadway network. So the model is a heuristic one, currently not yet validate. The paper is well articulated and developed, but generates some perplexity.

A1 The authors would like to thank Reviewer 3 for the comments that helped us to improve the quality of the manuscript. Please check also our comments to Reviewers 1 and 2. As also we responded to the comments of Reviewer 1 and 2, we would like to emphasise that our approach, except for the risk related to see walls is not heuristic, instead it is based on analogs and proportionalities for calculating the probability of occurrence of the events of a given magnitude/intensity which is assessed based on quantitative data, as most of the procedures for risk assessment are based on analogs.

Q2. First of all, the referee #3 agree with all the specific comments on the submitted paper from referees #1 and 2. One main concern is about the use of the velocity concept in several part of the text and particularly in the section about the slow landslides. Although velocity is widely adopted in many landslide classification systems, it is well known (from Physics) that in a force system the velocity of a rigid body mass is not representative of its equilibrium or disequilibrium state, which is demanded to the first derivative or gradient of the velocity vs time. About the classification and risk management of the "slow landslides", the main references to this kind of landslide are the monthly slope deformation (periodically measured in inclinometers) andthe one year cumulate displacements. So the Authors indirectly and correctly use the terms that contribute to the velocity vs time gradient (the acceleration).

A2. We use two criteria, the maximum monthly horizontal displacement rate and the annual cumulative horizontal displacement. The reason for using the monthly velocity (displacement rate) here is not to indicate the slope equilibrium or disequilibrium state, but to provide a measurement of the stresses and strains induced to the road due to the landslide movement, and to associate this movement with a certain extent of damage. Higher displacement rates (irrespectively of the landslide being in a dynamic equilibrium or not) induce larger actions on the road and cause higher damage. The probability of a given landslide intensity, based on displacement rate and the cumulative displacement is based on observations of the inclinometers and not on a geotechnical analysis providing results on the equilibrium/disequilibrium state.

Q3. Some points are not clear or instill some doubts: We start with the consideration that number and position of the available instruments in each site are well positioned and representative of the deformative field of the whole landslide body; are the deformation readings in the inclinometers constantly and regularly (which is the frequency) performed? Do the deformations of the inclinometer allow durability of the measures? Bigger movements are usually at the head, but do we analyze the correct one phenomenology, i.e. a shallow instead of a most dangerous and deeper incoming failure? Finally: measurements normally refer to a pre-peak failure stage: the post peak behaviour with the typical range of strength reduction (and then the hazard magnitude) is strongly controlled by the dominant lithology, which has been missed throughout the decisional points of the whole proposed model. Lithology and soil plasticity are common factors in many heuristic hazard models. Similar consideration should be extended to short term vs long term groundwater variations.

A3. We would like to thank the reviewer for these comments. The selected inclinometers to measure the data used here are in most cases placed in the vicinity of the road, with a few exceptions. Local accelerations, if any, in areas not monitored by the inclinometers are indeed not captured. However it is not always the case that those local movements affect the roads, and thus of interest. After 2010, deformation readings are constantly taken, every 3 months. As some inclinometers are functioning about 15 years now, there have been some gaps in measurements, but before 2011 (mostly in 2008-2009). Indeed, some inclinometers become inoperative after suffering large deformations. The proposed procedure takes this into account through the criterion of annual cumulative horizontal displacement. The slow-moving landslides in the area are well identified and geological and geotechnical studies have been performed, for the identification of the fracture/sliding surface and the back analysis of the stability, at least in most cases. As mentioned in the text, in the area there are clay materials with a viscous behaviour. Inclinometers in principle are deeper that the identified sliding surfaces. There are exceptional suspicions for deeper sliding surfaces not monitored. . As mentioned in section 3.3, the relation between the inclinometer indications and

СЗ

the road damage was established for 24 points, out of which 20 yield consistent predictions (coherent results between movement rate and damage), only one presented damage overestimation and 3 presented damage underestimation (out of which one inclinometer was identified to have low reliability). The afore-mentioned indicate a high rate of consistency of inclinometers with the damage. In exceptional cases, further geotechnical studies would have to be performed to provide us with new data for the risk calculation. In any case, the authors agree that the inclinometer measurement related uncertainties that the Reviewer mentions might be present, and they point this out in the discussion session. However, despite uncertainties, the focus here has been on the identification of patterns of movements, especially during and after the critical 2011 and 2013 events, trying to see the potential of each landslide for a given intensity (displacement rate), instead of measuring movements with precisión, and this has been possible.

Q4. The model has been applied to a system spatially extended for about 10.000 sq kilo- metres and developing several hundreds of kilometres (see fig.1), whose health and wellness rely on a limited number of instrumentations and on periodical inspections; it is characterized by several geological frames and by a well-developed surface hydrography. The local control apparatus is aged about 16 years and does not allow real time measurements in spite of its expensiveness. In other words, the monitoring system installed and which is the main source of data on which the proposed model works can be considered somewhat obsolete. Today the large areas can be controlled by means of active or passive PS INSAR techniques (measurement return time about 6 days); inclinometers can be flanked by optical or capacitive TDR and other kind of devices able of continuous and real-time response. While the meticulous and complete care in the model deserves great appreciation, some perplexities already expressed, together with the considerations about the repeatability of the scenarios, significantly limit the usefulness of the model in future scenarios.

A4. The methodology was developed according to the available data in the area and

an update of the monitoring systems was not part of this work. The Authors are well aware of the technologies and studies involving InSAR and DinSAR measurements for the calculation of the ground terrain movements on roads and certainly consider that these methods could be proved useful for future studies in the area.

Q5. Technical corrections In fig.6, Patterns of movement for landslides responsive and not responsive to rainfall, nowhere is reported a numerical scale (also indicative) of time

A5. The Figure was replaced and the scale was added.

Please also note the supplement to this comment: https://www.nat-hazards-earth-syst-sci-discuss.net/nhess-2018-234/nhess-2018-234-AC3-supplement.pdf

---

## Short Comment (SC1) · 14 Nov 2018

Please find as a supplement to this comment the new revised version of the mansucript.

Please also note the supplement to this comment:
https://www.nat-hazards-earth-syst-sci-discuss.net/nhess-2018-234/nhess-2018-234-SC1-supplement.pdf

---

## Author Comment (AC4) · 14 Nov 2018

Please find as a supplement to this comment the new revised version of the mansucript.

Please also note the supplement to this comment:
https://www.nat-hazards-earth-syst-sci-discuss.net/nhess-2018-234/nhess-2018-234-AC4-supplement.pdf
* * *

---

## Referee Report (RR1)

Journal: NHESS

Title: Integrated risk assessment due to slope instabilities at the roadway network of Gipuzkoa, Basque Country

Author(s): Olga Mavrouli et al.

General comments

The paper has been positively reviewed in many aspects; however, some critical issues already highlighted in the previous revision still remain.

The paper presents a model of evaluation of multirisk in selected points of the road network of Gipuzkoa, Basque Country. The proposed model is largely based on pre-existing monitoring equipment and techniques; it also draws on databases of significant events, but often scarce and/or inhomogeneous. The last circumstance, which is considered by Authors and can be considered as usually , due to the difference between technical and administrative documentations in wide extent networks, in fact makes difficult to evaluate return time and statistics of events, and therefore the temporal hazard.

The transformation of the inhomogeneous and scarce data in a quantitative score is made up with criteria driven by the judgement, so the model is mainly a heuristic one, currently not yet validated. The term heuristic is not dysphemistic and in our case means not statistic, not deterministic and not AI based.

The main problems relate to the low moving landslides. In this case, and unlike what is made in rockfalls, there is no information on the geological and geotechnical model of the PoR; the homogeneity of the PoR is therefore only "symptomatic". There is total lack of information on clay plasticity in slow moving landslides.

Local velocities, where used, are evaluated through the displacement of the head of inclinometers (which are referred to the toe). What about the overall velocity and the thickness of the moving layer? (do we analyze the correct one phenomenology, i.e. a shallow instead of a most dangerous and deeper incoming failure?).

Groundwater and their short term vs. long term variations are ignored; sensitivity to rainfall comes from the history of measured displacements. So the sensitivity to the rains, the triggering factor, requires a long historical data collection, which itself characterizes the level of hazard affecting the PoR.

The proposed model solves with accuracy the problems deriving from the scenarios and the quantity and type of data available, but with some approximation from a purely scientific point of view. The meticulous and complete care in the proposed model deserves great appreciation, but it ranges at max levels of a technical report. Modern and more performing monitoring techniques and a validation possible only with long historical series of measures significantly limit the usefulness of the model.

Technical corrections

In references, page 28 line 29 authors are cited twice.

---

## Author Response (AR2)

Dear Editor and Reviewers,

We would like to thank you again for your insights that helped us to clarify aspects of our work and of the applicability of our method, and to improve the quality of the paper.

Please find in the following our point-to-point answers to your comments (in blue).

**Answer to the Editor**

Thank you very much again for the revision of your manuscript with the title "Integrated risk assessment due to slope instabilities at the roadway network of Gipuzkoa, Basque Country". The revised manuscript was reviewed by two reviewers. Both of the reviewers acknowledged the changes and improvements of the manuscript based on the suggestions of the reviewers reports. One of the reviewer has now no further comments, the second one suggested to highlight and make more explicit the limitations of the current approaches and data. The reviewer provided also a report. After reading the manuscript again, I agree with the referee and ask therefore for minor revision of the manuscript.
I look forward to receiving the revised version of your manuscript. I will review the revised version of your manuscript.

The manuscript was revised, according to the suggestions of the Reviewer, highlighting in the conclusions the limitations of the used approach and data, with some additional text (please see manuscript_changes.pdf). Moreover, we provide a point-to-point answer to the comments of the reviewer.

**Answer to the Reviewer**

The paper has been positively reviewed in many aspects; however, some critical issues already highlighted in the previous revision still remain.

The paper presents a model of evaluation of multirisk in selected points of the road network of Gipuzkoa, Basque Country. The proposed model is largely based on pre-existing monitoring equipment and techniques; it also draws on databases of significant events, but often scarce and/or inhomogeneous. The last circumstance, which is considered by Authors and can be considered as usually , due to the difference between technical and administrative documentations in wide extent networks, in fact makes difficult to evaluate return time and statistics of events, and therefore the temporal hazard.

The transformation of the inhomogeneous and scarce data in a quantitative score is made up with criteria driven by the judgement, so the model is mainly a heuristic one, currently not yet validated. The term heuristic is not dysphemistic and in our case means not statistic, not deterministic and not AI based.

As mentioned in our previous response, the approaches followed are based on analogs and proportionalities for calculating the probability of occurrence of the events of a given magnitude/intensity which is assessed based on quantitative data, even if the latter is scarce, and transparent as well as reproducible quantitative criteria.

Uncertainties, bias, sampling errors are indeed present and further highlighted in the conclusions. Systematic validation remains an issue, and mentioned in the conclusions as well.

The main problems relate to the slow moving landslides. In this case, and unlike what is made in rockfalls, there is no information on the geological and geotechnical model of the PoR; the homogeneity of the PoR is therefore only "symptomatic". There is total lack of information on clay plasticity in slow moving landslides.

Local velocities, where used, are evaluated through the displacement of the head of inclinometers (which are referred to the toe). What about the overall velocity and the thickness of the moving layer? (do we analyze the correct one phenomenology, i.e. a shallow instead of a most dangerous and deeper incoming failure?).

Groundwater and their short term vs. long term variations are ignored; sensitivity to rainfall comes from the history of measured displacements. So the sensitivity to the rains, the triggering factor, requires a long historical data collection, which itself characterizes the level of hazard affecting the PoR.

The establishment of the temporal probabilities of reactivation is evidence-based, upon landslide movement measurements. It does not directly account for the landslide mechanism and type, neither for the clay plasticity and resistance, but indirectly it does, identifying patterns of movements for each landslide. We distinguish landslides in those that are responsive to rainfall and those that are not. For the former, we identify patterns of movement in relation to rainfall. For the latter, as response to rainfall is not clear indicating that the landslide velocity is controlled by other factors rather than that, we penalize lack of information related to the movement drivers, attributing higher probability of reactivation with a certain intensity. The recurrence of such reactivations, where possible, was checked from inclinometer indications. We highlight in the conclusions that this is an approximation and not a direct calculation of the probability.

As mention in a previous answer to Reviewer3, the slow-moving landslides in the area are well identified and geological and geotechnical studies have been performed, for the identification of the fracture/sliding surface and the back analysis of the stability, at least in most cases. As mentioned in the text, in the area there are clay materials with a viscous behaviour. Inclinometers in principle are deeper that the identified sliding surfaces. Hence deeper sliding surfaces would be exceptional. Nevertheless, this requirement is outlined in the conclusions, as suggested.

Possible variations of the landslide velocity within the landslide body are acknowledged in page 20. However, as mentioned in section 3.3, the relation between the inclinometer indications and the road damage was established for 24 points, out of which 20 yield consistent predictions (coherent results between movement rate and damage), only one presented damage overestimation and 3 presented damage underestimation (out of which one inclinometer was identified to have low reliability). The afore-mentioned indicate a high rate of consistency of inclinometers with the damage, and this has been our starting point. In exceptional cases, further geotechnical studies would have to be performed to provide us with new data for the risk calculation. This is as well highlighted in the text.

The proposed model solves with accuracy the problems deriving from the scenarios and the quantity and type of data available, but with some approximation from a purely scientific point of view. The

meticulous and complete care in the proposed model deserves great appreciation, but it ranges at max levels of a technical report. Modern and more performing monitoring techniques and a validation possible only with long historical series of measures significantly limit the usefulness of the model.

As mentioned in the introduction the goal of this work was to serve as a basis for the prioritization of the risk in the PoR of the studies road network, given the available data. The Authors acknowledge that for the validation of the proposed method, and on a long term basis, further monitoring techniques with high temporal resolution and precision either for slow or for rapid movements (e.g. SAR techniques, or LiDAR monitoring) and extended and reliable groundwater measurements can be proved useful, and included this in the conclusions as a suggestion.

In references, page 28 line 29 authors are cited twice.

Corrected.

[revised manuscript text omitted]